# Meta-Adaptive Nonlinear Control:
# Theory and Algorithms

**Guanya Shi**[†], **Kamyar Azizzadenesheli**[‡], **Michael O'Connell**[†], **Soon-Jo Chung**[†], **Yisong Yue**[†]

[†]Caltech  [‡]Purdue University

{gshi,moc,sjchung,yyue}@caltech.edu, kamyar@purdue.edu

## Abstract

We present an online multi-task learning approach for adaptive nonlinear control, which we call Online Meta-Adaptive Control (OMAC). The goal is to control a nonlinear system subject to adversarial disturbance and unknown *environment-dependent* nonlinear dynamics, under the assumption that the environment-dependent dynamics can be well captured with some shared representation. Our approach is motivated by robot control, where a robotic system encounters a sequence of new environmental conditions that it must quickly adapt to. A key emphasis is to integrate online representation learning with established methods from control theory, in order to arrive at a unified framework that yields both control-theoretic and learning-theoretic guarantees. We provide instantiations of our approach under varying conditions, leading to the first non-asymptotic end-to-end convergence guarantee for multi-task nonlinear control. OMAC can also be integrated with deep representation learning. Experiments show that OMAC significantly outperforms conventional adaptive control approaches which do not learn the shared representation, in inverted pendulum and 6-DoF drone control tasks under varying wind conditions[1].

## 1 Introduction

One important goal in autonomy and artificial intelligence is to enable autonomous robots to learn from prior experience to quickly adapt to new tasks and environments. Examples abound in robotics, such as a drone flying in different wind conditions [1], a manipulator throwing varying objects [2], or a quadruped walking over changing terrains [3]. Though those examples provide encouraging empirical evidence, when designing such adaptive systems, two important theoretical challenges arise, as discussed below.

First, from a learning perspective, the system should be able to learn an "efficient" representation from prior tasks, thereby permitting faster future adaptation, which falls into the categories of representation learning or meta-learning. Recently, a line of work has shown theoretically that learning representations (in the standard supervised setting) can significantly reduce sample complexity on new tasks [4–6]. Empirically, deep representation learning or meta-learning has achieved success in many applications [7], including control, in the context of meta-reinforcement learning [8–10]. However, theoretical benefits (in the end-to-end sense) of representation learning or meta-learning for adaptive control remain unclear.

Second, from a control perspective, the agent should be able to handle parametric model uncertainties with control-theoretic guarantees such as stability and tracking error convergence, which is a common adaptive control problem [11, 12]. For classic adaptive control algorithms, theoretical analysis often involves the use of Lyapunov stability and asymptotic convergence [11, 12]. Moreover, many recent

---

[1]Code and video: https://github.com/GuanyaShi/Online-Meta-Adaptive-Control

35th Conference on Neural Information Processing Systems (NeurIPS 2021).

studies aim to integrate ideas from learning, optimization, and control theory to design and analyze adaptive controllers using learning-theoretic metrics. Typical results guarantee non-asymptotic convergence in finite time horizons, such as regret [13–17] and dynamic regret [18–20]. However, these results focus on a single environment or task. A multi-task extension, especially whether and how prior experience could benefit the adaptation in new tasks, remains an open problem.

**Main contributions.** In this paper, we address both learning and control challenges in a unified framework and provide end-to-end guarantees. We derive a new method of Online Meta-Adaptive Control (OMAC) that controls uncertain nonlinear systems under a sequence of new environmental conditions. The underlying assumption is that the environment-dependent unknown dynamics can well be captured by a shared representation, which OMAC learns using a *meta-adapter*. OMAC then performs environment-specific updates using an *inner-adapter*.

We provide different instantiations of OMAC under varying assumptions and conditions. In the jointly and element-wise convex cases, we show sublinear cumulative control error bounds, which to our knowledge is the first non-asymptotic convergence result for multi-task nonlinear control. Compared to standard adaptive control approaches that do not have a meta-adapter, we show that OMAC possesses both stronger guarantees and empirical performance. We finally show how to integrate OMAC with deep representation learning, which further improves empirical performance.

## 2 Problem statement

We consider the setting where a controller encounters a sequence of $N$ environments, with each environment lasting $T$ time steps. We use *outer iteration* to refer to the iterating over the $N$ environments, and *inner iteration* to refer to the $T$ time steps within an environment.

**Notations:** We use superscripts (e.g., $(i)$ in $x_t^{(i)}$) to denote the index of the outer iteration where $1 \leq i \leq N$, and subscripts (e.g., $t$ in $x_t^{(i)}$) to denote the time index of the inner iteration where $1 \leq t \leq T$. We use $\texttt{step}(i, t)$ to refer to the inner time step $t$ at the $i^{\text{th}}$ outer iteration. $\| \cdot \|$ denotes the 2-norm of a vector or the spectral norm of a matrix. $\| \cdot \|_F$ denotes the Frobenius norm of a matrix and $\lambda_{\min}(\cdot)$ denotes the minimum eigenvalue of a real symmetric matrix. $\text{vec}(\cdot) \in \mathbb{R}^{mn}$ denotes the vectorization of a $m \times n$ matrix, and $\otimes$ denotes the Kronecker product. Finally, we use $u_{1:t}$ to denote a sequence $\{u_1, u_2, \cdots, u_t\}$.

We consider a discrete-time nonlinear control-affine system [21, 22] with environment-dependent uncertainty $f(x, c)$. The dynamics model at the $i^{\text{th}}$ outer iteration is:

$$x_{t+1}^{(i)} = f_0(x_t^{(i)}) + B(x_t^{(i)})u_t^{(i)} - f(x_t^{(i)}, c^{(i)}) + w_t^{(i)}, \quad 1 \leq t \leq T, \tag{1}$$

where the state $x_t^{(i)} \in \mathbb{R}^n$, the control $u_t^{(i)} \in \mathbb{R}^m$, $f_0 : \mathbb{R}^n \to \mathbb{R}^n$ is a known nominal dynamics model, $B : \mathbb{R}^n \to \mathbb{R}^{n \times m}$ is a known state-dependent actuation matrix, $c^{(i)} \subset \mathbb{R}^h$ is the unknown parameter that indicates an environmental condition, $f : \mathbb{R}^n \times \mathbb{R}^h \to \mathbb{R}^n$ is the unknown $c^{(i)}$-dependent dynamics model, and $w_t^{(i)}$ is disturbance, potentially adversarial. For simplicity we define $B_t^{(i)} = B(x_t^{(i)})$ and $f_t^{(i)} = f(x_t^{(i)}, c^{(i)})$.

**Interaction protocol.** We study the following adaptive nonlinear control problem under $N$ unknown environments. At the beginning of outer iteration $i$, the environment first selects $c^{(i)}$ (adaptively and adversarially), which is unknown to the controller, and then the controller makes decision $u_{1:T}^{(i)}$ under unknown dynamics $f(x_t^{(i)}, c^{(i)})$ and potentially adversarial disturbances $w_t^{(i)}$. To summarize:

1. **Outer iteration $i$.** A policy encounters environment $i$ ($i \in \{1, \ldots, N\}$), associated with unobserved variable $c^{(i)}$ (e.g., the wind condition for a flying drone). Run inner loop (Step 2).

2. **Inner loop.** Policy interacts with environment $i$ for $T$ time steps, observing $x_t^{(i)}$ and taking action $u_t^{(i)}$, with state/action dynamics following (1).

3. Policy optionally observes $c^{(i)}$ at the end of the inner loop (used for some variants of the analysis).

4. Increment $i = i + 1$ and repeat from Step 1.

We use average control error (ACE) as our performance metric:

**Definition 1** (Average control error). The average control error (ACE) of $N$ outer iterations (i.e., $N$ environments) with each lasting $T$ time steps, is defined as $\mathsf{ACE} = \frac{1}{TN} \sum_{i=1}^{N} \sum_{t=1}^{T} \|x_t^{(i)}\|$.

ACE can be viewed as a non-asymptotic generalization of steady-state error in control [23]. We make the following assumptions on the actuation matrix $B$, the nominal dynamics $f_0$, and disturbance $w_t^{(i)}$:

**Assumption 1** (Full actuation, bounded disturbance, and e-ISS assumptions). *We consider fully-actuated systems, i.e., for all $x$, $\mathrm{rank}(B(x)) = n$. $\|w_t^{(i)}\| \leq W, \forall t, i$. Moreover, the nominal dynamics $f_0$ is exponentially input-to-state stable (e-ISS): let constants $\beta, \gamma \geq 0$ and $0 \leq \rho < 1$. For a sequence $v_{1:t-1} \in \mathbb{R}^n$, consider the dynamics $x_{k+1} = f_0(x_k) + v_k, 1 \leq k \leq t-1$. $x_t$ satisfies:*

$$\|x_t\| \leq \beta \rho^{t-1} \|x_1\| + \gamma \sum_{k=1}^{t-1} \rho^{t-1-k} \|v_k\|. \tag{2}$$

With the e-ISS property in Assumption 1, we have the following bound that connects ACE with the average squared loss between $B_t^{(i)} u_t^{(i)} + w_t^{(i)}$ and $f_t^{(i)}$.

**Lemma 1.** *Assume $x_1^{(i)} = 0, \forall i$. The average control error (ACE) is bounded as:*

$$\frac{\sum_{i=1}^{N} \sum_{t=1}^{T} \|x_t^{(i)}\|}{TN} \leq \frac{\gamma}{1-\rho} \sqrt{\frac{\sum_{i=1}^{N} \sum_{t=1}^{T} \|B_t^{(i)} u_t^{(i)} - f_t^{(i)} + w_t^{(i)}\|^2}{TN}}. \tag{3}$$

The proof can be found in the Appendix A.1. We assume $x_1^{(i)} = 0$ for simplicity: the influence of non-zero and bounded $x_1^{(i)}$ is a constant term in each outer iteration, from the e-ISS property (2).

**Remark on the e-ISS assumption and the ACE metric.** Note that an exponentially stable linear system $f_0(x_t) = Ax_t$ (i.e., the spectral radius of $A$ is $< 1$) satisfies the exponential ISS (e-ISS) assumption. However, in nonlinear systems e-ISS is a stronger assumption than exponential stability. For both linear and nonlinear systems, the e-ISS property of $f_0$ is usually achieved by applying some stable feedback controller to the system[2], i.e., $f_0$ is the closed-loop dynamics [11, 24, 16]. e-ISS assumption is standard in both online adaptive linear control [24, 16, 14] and nonlinear control [13], and practical in robotic control such as drones [25]. In ACE we consider a regulation task, but it can also capture trajectory tracking task with time-variant nominal dynamics $f_0$ under incremental stability assumptions [13]. We only consider the regulation task in this paper for simplicity.

**Generality.** We would like to emphasize the generality of our dynamics model (1). The nominal control-affine part can model general fully-actuated robotic systems via Euler-Langrange equations [21, 22], and the unknown part $f(x, c)$ is nonlinear in $x$ and $c$. We only need to assume the disturbance $w_t^{(i)}$ is bounded, which is more general than stochastic settings in linear [14–16] and nonlinear [13] cases. For example, $w_t^{(i)}$ can model extra $(x, u, c)$-dependent uncertainties or adversarial disturbances. Moreover, the environment sequence $c^{(1:N)}$ could also be adversarial. In term of the extension to under-actuated systems, all the results in this paper hold for the *matched uncertainty* setting [13], i.e., in the form $x_{t+1}^{(i)} = f_0(x_t^{(i)}) + B(x_t^{(i)})(u_t^{(i)} - f(x_t^{(i)}, c^{(i)})) + w_t^{(i)}$ where $B(x_t^{(i)})$ is not necessarily full rank (e.g., drone and inverted pendulum experiments in Section 5). Generalizing to other under-actuated systems is interesting future work.

## 3 Online meta-adaptive control (OMAC) algorithm

The design of our online meta-adaptive control (OMAC) approach comprises four pieces: the policy class, the function class, the inner loop (within environment) adaptation algorithm $\mathcal{A}_2$, and the outer loop (between environment) online learning algorithm $\mathcal{A}_1$.

**Policy class.** We focus on the class of certainty-equivalence controllers [13, 26, 14], which is a general class of model-based controllers that also includes linear feedback controllers commonly

---

[2]For example, consider $x_{t+1} = \frac{3}{2} x_t + 2 \sin x_t + \bar{u}_t$. With a feedback controller $\bar{u}_t = u_t - x_t - 2 \sin x_t$, the closed-loop dynamics is $x_{t+1} = \frac{1}{2} x_t + u_t$, where $f_0(x) = \frac{1}{2} x$ is e-ISS.

studied in online control [26, 14]. After a model is learned from past data, a controller is designed by treating the learned model as the truth [12]. Formally, at $\mathtt{step}(i,t)$, the controller first estimates $\hat{f}_t^{(i)}$ (an estimation of $f_t^{(i)}$) based on past data, and then executes $u_t^{(i)} = B_t^{(i)\dagger}\hat{f}_t^{(i)}$, where $(\cdot)^\dagger$ is the pseudo inverse. Note that from Lemma 1, the average control error of the *omniscient controller* using $\hat{f}_t^{(i)} = f_t^{(i)}$ (i.e., the controller perfectly knows $f(x,c)$) is upper bounded as[3]:

$$\mathsf{ACE}(\text{omniscient}) \leq \gamma W/(1-\rho).$$

$\mathsf{ACE}(\text{omniscient})$ can be viewed as a fundamental limit of the certainty-equivalence policy class.

**Function class $F$ for representation learning.** In OMAC, we need to define a function class $F(\phi(x;\hat{\Theta}),\hat{c})$ to compute $\hat{f}_t^{(i)}$ (i.e., $\hat{f}_t^{(i)} = F(\phi(x_t^{(i)};\hat{\Theta}),\hat{c})$), where $\phi$ (parameterized by $\hat{\Theta}$) is a representation shared by all environments, and $\hat{c}$ is an environment-specific latent state. From a theoretical perspective, the main consideration of the choice of $F(\phi(x),\hat{c})$ is on how it effects the resulting learning objective. For instance, $\phi$ represented by a Deep Neural Network (DNN) would lead to a highly non-convex learning objective. In this paper, we focus on the setting $\hat{\Theta} \in \mathbb{R}^p, \hat{c} \in \mathbb{R}^h$, and $p \gg h$, i.e., it is much more expensive to learn the shared representation $\phi$ (e.g., a DNN) than "fine-tuning" via $\hat{c}$ in a specific environment, which is consistent with meta-learning [9] and representation learning [4, 7] practices.

**Inner loop adaptive control.** We take a modular approach in our algorithm design, in order to cleanly leverage state-of-the-art methods from online learning, representation learning, and adaptive control. When interacting with a single environment (for $T$ time steps), we keep the learned representation $\phi$ fixed, and use that representation for adaptive control by treating $\hat{c}$ as an unknown low-dimensional parameter. We can utilize any adaptive control method such as online gradient descent, velocity gradient, or composite adaptation [27, 13, 11, 1].

**Outer loop online learning.** We treat the outer loop (which iterates between environments) as an online learning problem, where the goal is to learn the shared representation $\phi$ that optimizes the inner loop adaptation performance. Theoretically, we can reason about the analysis by setting up a hierarchical or nested online learning procedure (adaptive control nested within online learning).

**Design goal.** Our goal is to design a meta-adaptive controller that has low ACE, ideally converging to $\mathsf{ACE}(\text{omniscient})$ as $T, N \rightarrow \infty$. In other words, OMAC should converge to performing as good as the omniscient controller with perfect knowledge of $f(x,c)$.

Algorithm 1 describes the OMAC algorithm. Since $\phi$ is environment-invariant and $p \gg h$, we only adapt $\hat{\Theta}$ at the end of each outer iteration. On the other hand, because $c^{(i)}$ varies in different environments, we adapt $\hat{c}$ at each inner step. As shown in Algorithm 1, at $\mathtt{step}(i,t)$, after applying $u_t^{(i)}$, the controller observes the next state $x_{t+1}^{(i)}$ and computes: $y_t^{(i)} \triangleq f_0(x_t^{(i)}) + B_t^{(i)}u_t^{(i)} - x_{t+1}^{(i)} = f_t^{(i)} - w_t^{(i)}$, which is a disturbed measurement of the ground truth $f_t^{(i)}$. We then define $\ell_t^{(i)}(\hat{\Theta},\hat{c}) \triangleq \|F(\phi(x_t^{(i)};\hat{\Theta}),\hat{c}) - y_t^{(i)}\|^2$ as the observed loss at $\mathtt{step}(i,t)$, which is a squared loss between the disturbed measurement of $f_t^{(i)}$ and the model prediction $F(\phi(x_t^{(i)};\hat{\Theta}),\hat{c})$.

**Instantiations.** Depending on $\{F(\phi(x;\hat{\Theta}),\hat{c}), \mathcal{A}_1, \mathcal{A}_2, \mathtt{ObserveEnv}\}$, we consider three settings:

- **Convex case** (Section 4.1): The observed loss $\ell_t^{(i)}$ is convex with respect to $\hat{\Theta}$ and $\hat{c}$.

- **Element-wise convex case** (Section 4.2): Fixing $\hat{\Theta}$ or $\hat{c}$, $\ell_t^{(i)}$ is convex with respect to the other.

- **Deep learning case** (Section 5): In this case, we use a DNN with weight $\hat{\Theta}$ to represent $\phi$.

## 4 Main theoretical results

### 4.1 Convex case

In this subsection, we focus on a setting where the observed loss $\ell_t^{(i)}(\hat{\Theta},\hat{c}) = \|F(\phi(x;\hat{\Theta}),\hat{c}) - y_t^{(i)}\|^2$ is convex with respect to $\hat{\Theta}$ and $\hat{c}$. We provide the following example to illustrate this case and highlight its difference between conventional adaptive control (e.g., [13, 12]).

---

[3]This upper bound is tight. Consider a scalar system $x_{t+1} = ax_t + u_t - f(x_t) + w$ with $|a| < 1$ and $w$ a constant. In this case $\rho = |a|, \gamma = 1$, and the omniscient controller $u_t = f(x_t)$ yields $\mathsf{ACE} = \gamma|w|/(1-\rho)$.

**Algorithm 1:** Online Meta-Adaptive Control (OMAC) algorithm

**Input:** Meta-adapter $\mathcal{A}_1$; inner-adapter $\mathcal{A}_2$; model $F(\phi(x;\hat{\Theta}),\hat{c})$; Boolean `ObserveEnv`

**for** $i = 1, \cdots, N$ **do**

    The environment selects $c^{(i)}$

    **for** $t = 1, \cdots, T$ **do**

        Compute $\hat{f}_t^{(i)} = F(\phi(x_t^{(i)};\hat{\Theta}^{(i)}),\hat{c}_t^{(i)})$

        Execute $u_t^{(i)} = B_t^{(i)\dagger}\hat{f}_t^{(i)}$   `//certainty-equivalence policy`

        Observe $x_{t+1}^{(i)}, y_t^{(i)} = f(x_t^{(i)}, c^{(i)}) - w_t^{(i)}$, and $\ell_t^{(i)}(\hat{\Theta},\hat{c}) = \|F(\phi(x_t^{(i)};\hat{\Theta}),\hat{c}) - y_t^{(i)}\|^2$

            `//`$y_t^{(i)}$` is a noisy measurement of `$f$` and `$\ell_t^{(i)}$` is the observed loss`

        Construct an inner cost function $g_t^{(i)}(\hat{c})$ by $\mathcal{A}_2$   `//`$g_t^{(i)}$` is a function of `$\hat{c}$

        Inner-adaptation: $\hat{c}_{t+1}^{(i)} \leftarrow \mathcal{A}_2(\hat{c}_t^{(i)}, g_{1:t}^{(i)})$

    **if** `ObserveEnv` **then** Observe $c^{(i)}$   `//only used in some instantiations`

    Construct an outer cost function $G^{(i)}(\hat{\Theta})$ by $\mathcal{A}_1$   `//`$G^{(i)}$` is a function of `$\hat{\Theta}$

    Meta-adaptation: $\hat{\Theta}^{(i+1)} \leftarrow \mathcal{A}_1(\hat{\Theta}^{(i)}, G^{(1:i)})$

**Example 1.** We consider a model $F(\phi(x;\hat{\Theta}),\hat{c}) = Y_1(x)\hat{\Theta} + Y_2(x)\hat{c}$ to estimate $f$:

$$\hat{f}_t^{(i)} = Y_1(x_t^{(i)})\hat{\Theta}^{(i)} + Y_2(x_t^{(i)})\hat{c}_t^{(i)}, \tag{4}$$

where $Y_1 : \mathbb{R}^n \to \mathbb{R}^{n \times p}, Y_2 : \mathbb{R}^n \to \mathbb{R}^{n \times h}$ are two known bases. Note that conventional adaptive control approaches typically concatenate $\hat{\Theta}$ and $\hat{c}$ and adapt on both at each time step, regardless of the environment changes (e.g., [13]). Since $p \gg h$, such concatenation is computationally much more expensive than OMAC, which only adapts $\hat{\Theta}$ in outer iterations.

Because $\ell_t^{(i)}(\hat{\Theta},\hat{c})$ is *jointly* convex with respective to $\hat{\Theta}$ and $\hat{c}$, the OMAC algorithm in this case falls into the category of Nested Online Convex Optimization (Nested OCO) [28]. The choice of $g_t^{(i)}, G^{(i)}, \mathcal{A}_1, \mathcal{A}_2$ and `ObserveEnv` are depicted in Table 1. Note that in the convex case OMAC does not need to know $c^{(i)}$ in the whole process (`ObserveEnv` = False).

| | |
|---|---|
| $F(\phi(x;\hat{\Theta}),\hat{c})$ | Any $F$ model such that $\ell_t^{(i)}(\hat{\Theta},\hat{c})$ is convex (e.g., Example 1) |
| $g_t^{(i)}(\hat{c})$ | $\nabla_{\hat{c}}\ell_t^{(i)}(\hat{\Theta}^{(i)},\hat{c}_t^{(i)}) \cdot \hat{c}$ |
| $G^{(i)}(\hat{\Theta})$ | $\sum_{t=1}^T \nabla_{\hat{\Theta}}\ell_t^{(i)}(\hat{\Theta}^{(i)},\hat{c}_t^{(i)}) \cdot \hat{\Theta}$ |
| $\mathcal{A}_1$ | With a convex set $\mathcal{K}_1$, $\mathcal{A}_1$ initializes $\hat{\Theta}^{(1)} \in \mathcal{K}_1$ and returns $\hat{\Theta}^{(i+1)} \in \mathcal{K}_1, \forall i$. $\mathcal{A}_1$ has sublinear regret, i.e., the total regret of $\mathcal{A}_1$ is $T \cdot o(N)$ (e.g., online gradient descent) |
| $\mathcal{A}_2$ | With a convex set $\mathcal{K}_2$, $\forall i$, $\mathcal{A}_2$ initializes $\hat{c}_1^{(i)} \in \mathcal{K}_2$ and returns $\hat{c}_{t+1}^{(i)} \in \mathcal{K}_2, \forall t$. $\mathcal{A}_2$ has sublinear regret, i.e., the total regret of $\mathcal{A}_2$ is $N \cdot o(T)$ (e.g., online gradient descent) |
| `ObserveEnv` | False |

Table 1: OMAC with convex loss

As shown in Table 1, at the end of $\text{step}(i,t)$ we fix $\hat{\Theta} = \hat{\Theta}^{(i)}$ and update $\hat{c}_{t+1}^{(i)} \in \mathcal{K}_2$ using $\mathcal{A}_2(\hat{c}_t^{(i)}, g_{1:t}^{(i)})$, which is an OCO problem with linear costs $g_{1:t}^{(i)}$. On the other hand, at the end of outer iteration $i$, we update $\hat{\Theta}^{(i+1)} \in \mathcal{K}_1$ using $\mathcal{A}_1(\hat{\Theta}^{(i)}, G^{(1:i)})$, which is another OCO problem with linear costs $G^{(1:i)}$. From [28], we have the following regret relationship:

**Lemma 2** (Nested OCO regret bound, [28]). *OMAC (Algorithm 1) specified by Table 1 has regret:*

$$\sum_{i=1}^N \sum_{t=1}^T \ell_t^{(i)}(\hat{\Theta}^{(i)},\hat{c}_t^{(i)}) - \min_{\Theta \in \mathcal{K}_1} \sum_{i=1}^N \min_{c^{(i)} \in \mathcal{K}_2} \sum_{t=1}^T \ell_t^{(i)}(\Theta, c^{(i)})$$

$$\leq \underbrace{\sum_{i=1}^N G^{(i)}(\hat{\Theta}^{(i)}) - \min_{\Theta \in \mathcal{K}_1} \sum_{i=1}^N G^{(i)}(\Theta)}_{\text{the total regret of } \mathcal{A}_1, T \cdot o(N)} + \underbrace{\sum_{i=1}^N \sum_{t=1}^T g_t^{(i)}(\hat{c}_t^{(i)}) - \sum_{i=1}^N \min_{c^{(i)} \in \mathcal{K}_2} \sum_{t=1}^T g_t^{(i)}(c^{(i)})}_{\text{the total regret of } \mathcal{A}_2, N \cdot o(T)}. \tag{5}$$

Note that the total regret of $\mathcal{A}_1$ is $T \cdot o(N)$ because $G^{(i)}$ is scaled up by a factor of $T$. With Lemmas 1 and 2, we have the following guarantee for the average control error.

**Theorem 3** (OMAC ACE bound with convex loss). *Assume the unknown dynamics model is $f(x, c) = F(\phi(x; \Theta), c)$. Assume the true parameters $\Theta \in \mathcal{K}_1$ and $c^{(i)} \in \mathcal{K}_2, \forall i$. Then OMAC (Algorithm 1) specified by Table 1 ensures the following* ACE *guarantee:*

$$\mathsf{ACE} \leq \frac{\gamma}{1 - \rho} \sqrt{W^2 + \frac{o(T)}{T} + \frac{o(N)}{N}}.$$

To further understand Theorem 3 and compare OMAC with conventional adaptive control approaches, we provide the following corollary using the model in Example 1.

**Corollary 4.** *Suppose the unknown dynamics model is $f(x, c) = Y_1(x)\Theta + Y_2(x)c$, where $Y_1 : \mathbb{R}^n \to \mathbb{R}^{n \times p}, Y_2 : \mathbb{R}^n \to \mathbb{R}^{n \times h}$ are known bases. We assume $\|\Theta\| \leq K_\Theta$ and $\|c^{(i)}\| \leq K_c, \forall i$. Moreover, we assume $\|Y_1(x)\| \leq K_1$ and $\|Y_2(x)\| \leq K_2, \forall x$. In Table 1 we use $\hat{f}_t^{(i)} = Y_1(x_t^{(i)})\hat{\Theta}^{(i)} + Y_2(x_t^{(i)})\hat{c}_t^{(i)}$, and Online Gradient Descent (OGD) [27] for both $\mathcal{A}_1$ and $\mathcal{A}_2$, with learning rates $\bar{\eta}^{(i)}$ and $\eta_t^{(i)}$ respectively. We set $\mathcal{K}_1 = \{\hat{\Theta} : \|\hat{\Theta}\| \leq K_\Theta\}$ and $\mathcal{K}_2 = \{\hat{c} : \|\hat{c}\| \leq K_c\}$. If we schedule the learning rates as:*

$$\bar{\eta}^{(i)} = \frac{2K_\Theta}{\underbrace{(4K_1^2 K_\Theta + 4K_1 K_2 K_c + 2K_1 W)}_{C_1}T\sqrt{i}}, \quad \eta_t^{(i)} = \frac{2K_c}{\underbrace{(4K_2^2 K_c + 4K_1 K_2 K_\Theta + 2K_2 W)}_{C_2}\sqrt{t}},$$

*then the average control performance is bounded as:*

$$\mathsf{ACE(OMAC)} \leq \frac{\gamma}{1 - \rho} \sqrt{W^2 + 3\left(K_\Theta C_1 \frac{1}{\sqrt{N}} + K_c C_2 \frac{1}{\sqrt{T}}\right)}.$$

*Moreover, the baseline adaptive control which uses OGD to adapt $\hat{\Theta}$ and $\hat{c}$ at each time step satisfies:*

$$\mathsf{ACE(\text{baseline adaptive control})} \leq \frac{\gamma}{1 - \rho} \sqrt{W^2 + 3\sqrt{K_\Theta^2 + K_c^2}\sqrt{C_1^2 + C_2^2}\frac{1}{\sqrt{T}}}.$$

Note that $\mathsf{ACE}(\text{baseline adaptive control})$ does not improve as $N$ increases (i.e., encountering more environments has no benefit). If $p \gg h$, we potentially have $K_1 \gg K_2$ and $K_\Theta \gg K_c$, so $C_1 \gg C_2$. Therefore, the ACE upper bound of OMAC is better than the baseline adaptation if $N > T$, which is consistent with recent representation learning results [4, 5]. It is also worth noting that the baseline adaptation is much more computationally expensive, because it needs to adapt both $\hat{\Theta}$ and $\hat{c}$ at each time step. Intuitively, OMAC improves convergence because the meta-adapter $\mathcal{A}_1$ approximates the environment-invariant part $Y_1(x)\Theta$, which makes the inner-adapter $\mathcal{A}_2$ much more efficient.

## 4.2 Element-wise convex case

In this subsection, we consider the setting where the loss function $\ell_t^{(i)}(\hat{\Theta}, \hat{c})$ is element-wise convex with respect to $\hat{\Theta}$ and $\hat{c}$, i.e., when one of $\hat{\Theta}$ or $\hat{c}$ is fixed, $\ell_t^{(i)}$ is convex with respect to the other one. For instance, $F$ could be function as depicted in Example 2. Outside the context of control, such bilinear models are also studied in representation learning [4, 5] and factorization bandits [29, 30].

**Example 2.** We consider a model $F(\phi(x; \hat{\Theta}), \hat{c}) = Y(x)\hat{\Theta}\hat{c}$ to estimate $f$:

$$\hat{f}_t^{(i)} = Y(x_t^{(i)})\hat{\Theta}^{(i)}\hat{c}_t^{(i)}, \tag{6}$$

where $Y : \mathbb{R}^n \to \mathbb{R}^{n \times \bar{p}}$ is a known basis, $\hat{\Theta}^{(i)} \in \mathbb{R}^{\bar{p} \times h}$, and $\hat{c}_t^{(i)} \in \mathbb{R}^h$. Note that the dimensionality of $\hat{\Theta}$ is $p = \bar{p}h$. Conventional adaptive control typically views $\hat{\Theta}\hat{c}$ as a vector in $\mathbb{R}^{\bar{p}}$ and adapts it at each time step regardless of the environment changes [13].

Compared to Section 4.1, the element-wise convex case poses new challenges: i) the coupling between $\hat{\Theta}$ and $\hat{c}$ brings inherent non-identifiability issues; and ii) statistical learning guarantees

typical need i.i.d. assumptions on $c^{(i)}$ and $x_t^{(i)}$ [4, 5]. These challenges are further amplified by the underlying unknown nonlinear system (1). Therefore in this section we set $\texttt{ObserveEnv} = \text{True}$, i.e., the controller has access to the true environmental condition $c^{(i)}$ at the end of the $i^{\text{th}}$ outer iteration, which is practical in many systems when $c^{(i)}$ has a concrete physical meaning, e.g., drones with wind disturbances [1, 31].

| | |
|---|---|
| $F(\phi(x;\hat{\Theta}),\hat{c})$ | Any $F$ model such that $\ell_t^{(i)}(\hat{\Theta},\hat{c})$ is element-wise convex (e.g., Example 2) |
| $g_t^{(i)}(\hat{c})$ | $\ell_t^{(i)}(\hat{\Theta}^{(i)},\hat{c})$ |
| $G^{(i)}(\hat{\Theta})$ | $\sum_{t=1}^{T} \ell_t^{(i)}(\hat{\Theta}, c^{(i)})$ |
| $\mathcal{A}_1$ & $\mathcal{A}_2$ | Same as Table 1 |
| $\texttt{ObserveEnv}$ | True |

Table 2: OMAC with element-wise convex loss

The inputs to OMAC for the element-wise convex case are specified in Table 2. Compared to the convex case in Table 1, difference includes i) the cost functions $g_t^{(i)}$ and $G^{(i)}$ are convex, not necessarily linear; and ii) since $\texttt{ObserveEnv} = \text{True}$, in $G^{(i)}$ we use the true environmental condition $c^{(i)}$. With inputs specified in Table 2, Algorithm 1 has ACE guarantees in the following theorem.

**Theorem 5** (OMAC ACE bound with element-wise convex loss). *Assume the unknown dynamics model is $f(x,c) = F(\phi(x;\Theta),c)$. Assume the true parameter $\Theta \in \mathcal{K}_1$ and $c^{(i)} \in \mathcal{K}_2, \forall i$. Then OMAC (Algorithm 1) specified by Table 2 ensures the following $\mathsf{ACE}$ guarantee:*

$$\mathsf{ACE} \leq \frac{\gamma}{1-\rho}\sqrt{W^2 + \frac{o(T)}{T} + \frac{o(N)}{N}}.$$

### 4.2.1 Faster convergence with sub Gaussian and environment diversity assumptions

Since the cost functions $g_t^{(i)}$ and $G^{(i)}$ in Table 2 are not necessarily strongly convex, the ACE upper bound in Theorem 5 is typically $\frac{\gamma}{1-\rho}\sqrt{W^2 + O(1/\sqrt{T}) + O(1/\sqrt{N})}$ (e.g., using OGD for both $\mathcal{A}_1$ and $\mathcal{A}_2$). However, for the bilinear model in Example 2, it is possible to achieve faster convergence with extra sub Gaussian and *environment diversity* assumptions.

| | |
|---|---|
| $F(\phi(x;\hat{\Theta}),\hat{c})$ | The bilinear model in Example 2 |
| $g_t^{(i)}(\hat{c})$ | $\ell_t^{(i)}(\hat{\Theta}^{(i)},\hat{c})$ |
| $G^{(i)}(\hat{\Theta})$ | $\lambda\|\hat{\Theta}\|_F^2 + \sum_{j=1}^{i}\sum_{t=1}^{T} \ell_t^{(j)}(\hat{\Theta}, c^{(j)})$ with some $\lambda > 0$ |
| $\mathcal{A}_1$ | $\hat{\Theta}^{(i+1)} = \arg\min_{\hat{\Theta}} G^{(i)}(\hat{\Theta})$ (i.e., Ridge regression) |
| $\mathcal{A}_2$ | Same as Table 1 |
| $\texttt{ObserveEnv}$ | True |

Table 3: OMAC with bilinear model

With a bilinear model, the inputs to the OMAC algorithm are shown in Table 3. With extra assumptions on $w_t^{(i)}$ and the environment, we have the following ACE guarantees.

**Theorem 6** (OMAC ACE bound with bilinear model). *Consider an unknown dynamics model $f(x,c) = Y(x)\Theta c$ where $Y : \mathbb{R}^n \rightarrow \mathbb{R}^{n\times\bar{p}}$ is a known basis and $\Theta \in \mathbb{R}^{\bar{p}\times h}$. Assume each component of the disturbance $w_t^{(i)}$ is $R$-sub-Gaussian, the true parameters $\|\Theta\|_F \leq K_\Theta$, $\|c^{(i)}\| \leq K_c, \forall i$, and $\|Y(x)\|_F \leq K_Y, \forall x$. Define $Z_t^{(j)} = c^{(j)\top} \otimes Y(x_t^{(j)}) \in \mathbb{R}^{n\times\bar{p}h}$ and assume environment diversity: $\lambda_{\min}(\sum_{j=1}^{i}\sum_{t=1}^{T} Z_t^{(j)\top} Z_t^{(j)}) = \Omega(i)$. Then OMAC (Algorithm 1) specified by Table 3 has the following $\mathsf{ACE}$ guarantee (with probability at least $1-\delta$):*

$$\mathsf{ACE} \leq \frac{\gamma}{1-\rho}\sqrt{W^2 + \frac{o(T)}{T} + O\left(\frac{\log(NT/\delta)\log(N)}{N}\right)}. \tag{7}$$

*If we relax the environment diversity condition to $\Omega(\sqrt{i})$, the last term becomes $O(\frac{\log(NT/\delta)}{\sqrt{N}})$.*

The sub-Gaussian assumption is widely used in statistical learning theory to obtain concentration bounds [32, 4]. The environment diversity assumption states that $c^{(i)}$ provides "new information"

in every outer iteration such that the minimum eigenvalue of $\sum_{j=1}^{i} \sum_{t=1}^{T} Z_t^{(j)\top} Z_t^{(j)}$ grows linearly as $i$ increases. Note that we do not need $\lambda_{\min}(\sum_{j=1}^{i} \sum_{t=1}^{T} Z_t^{(j)\top} Z_t^{(j)})$ to increase as $T$ goes up. Moreover, if we relax the condition to $\Omega(\sqrt{i})$, the ACE bound becomes worse than the general element-wise convex case (the last term is $O(1/\sqrt{N})$), which implies the importance of "linear" environment diversity $\Omega(i)$. Task diversity has been shown to be important for representation learning [4, 33]. We provide a proof sketch here and the full proof can be found in the Appendix A.6.

**Proof sketch.** In the outer loop we use the martingale concentration bound [32] and the environment diversity assumption to bound $\|\hat{\Theta}^{(i+1)} - \Theta\|_F^2 \leq O(\frac{\log(iT/\delta)}{i}), \forall i \geq 1$ with probability at least $1 - \delta$. Then, we use Lemma 1 to show how the outer loop concentration bound and the inner loop regret bound of $\mathcal{A}_2$ translate to ACE.

## 5 Deep OMAC and experiments

We now introduce deep OMAC, a deep representation learning based OMAC algorithm. Table 4 shows the instantiation. As shown in Table 4, Deep OMAC employs a DNN to represent $\phi$, and uses gradient descent for optimization. With the model[4] $\phi(x; \hat{\Theta}) \cdot \hat{c}$, the meta-adapter $\mathcal{A}_1$ updates the representation $\phi$ via deep learning, and the inner-adapter $\mathcal{A}_2$ updates a linear layer $\hat{c}$.

| | |
|---|---|
| $F(\phi(x; \hat{\Theta}), \hat{c})$ | $\phi(x; \hat{\Theta}) \cdot \hat{c}$, where $\phi(x; \hat{\Theta}) : \mathbb{R}^n \to \mathbb{R}^{n \times h}$ is a neural network with weight $\hat{\Theta}$ |
| $g_t^{(i)}(\hat{c})$ | $\ell_t^{(i)}(\hat{\Theta}^{(i)}, \hat{c})$ |
| $\mathcal{A}_1$ | Gradient descent: $\hat{\Theta}^{(i+1)} \leftarrow \hat{\Theta}^{(i)} - \eta \nabla_{\hat{\Theta}} \sum_{t=1}^{T} \ell_t^{(i)}(\hat{\Theta}, \hat{c}_t^{(i)})$ |
| $\mathcal{A}_2$ | Same as Table 1 |
| ObserveEnv | False |

Table 4: OMAC with deep learning

To demonstrate the performance of OMAC, we consider two sets of experiments:

- **Inverted pendulum with external wind, gravity mismatch, and unknown damping.** The continuous-time model is $ml^2\ddot{\theta} - ml\hat{g}\sin\theta = u + f(\theta, \dot{\theta}, c)$, where $\theta$ is the pendulum angle, $\dot{\theta}/\ddot{\theta}$ is the angular velocity/acceleration, $m$ is the pendulum mass and $l$ is the length, $\hat{g}$ is the gravity estimation, $c$ is the unknown parameter that indicates the external wind condition, and $f(\theta, \dot{\theta}, c)$ is the unknown dynamics which depends on $c$, but also includes $c$-invariant terms such as damping and gravity mismatch. This model generalizes [35] by considering damping and gravity mismatch.

- **6-DoF quadrotor with 3-D wind disturbances.** We consider a 6-DoF quadrotor model with unknown wind-dependent aerodynamic force $f(x, c) \in \mathbb{R}^3$, where $x \in \mathbb{R}^{13}$ is the drone state (including position, velocity, attitude, and angular velocity) and $c$ is the unknown parameter indicating the wind condition. We incorporate a realistic high-fidelity aerodynamic model from [36].

We consider 6 different controllers in the experiment (see more details about the dynamics model and controllers in the Appendix A.7):

- **No-adapt** is simply using $\hat{f}_t^{(i)} = 0$, and **omniscient** is using $\hat{f}_t^{(i)} = f_t^{(i)}$.

- **OMAC (convex)** is based on Example 1, where $\hat{f}_t^{(i)} = Y_1(x_t^{(i)})\hat{\Theta}^{(i)} + Y_2(x_t^{(i)})\hat{c}_t^{(i)}$. We use random Fourier features [37, 13] to generate both $Y_1$ and $Y_2$. We use OGD for both $\mathcal{A}_1$ and $\mathcal{A}_2$ in Table 1.

- **OMAC (bi-convex)** is based on Example 2, where $\hat{f}_t^{(i)} = Y(x_t^{(i)})\hat{\Theta}^{(i)}\hat{c}_t^{(i)}$. Similarly, we use random Fourier features to generate $Y$. Although the theoretical result in Section 4.2 requires ObserveEnv = True, we relax this in our experiments and use $G^{(i)}(\hat{\Theta}) = \sum_{t=1}^{T} \ell_t^{(i)}(\hat{\Theta}, \hat{c}_t^{(i)})$ in Table 2, instead of $\sum_{t=1}^{T} \ell_t^{(i)}(\hat{\Theta}, c^{(i)})$. We also deploy OGD for $\mathcal{A}_1$ and $\mathcal{A}_2$. **Baseline** has the same procedure except with $\hat{\Theta}^{(i)} \equiv \hat{\Theta}^{(1)}$, i.e., it calls the inner-adapter $\mathcal{A}_2$ at every step and does not update $\hat{\Theta}$, which is standard in adaptive control [13, 12].

---

[4]The intuition behind this structure is that, any analytic function $\bar{f}(x, \bar{c})$ can be approximated by $\phi(x)c(\bar{c})$ with a universal approximator $\phi$ [34]. We provide a detailed and theoretical justification in the Appendix A.7.

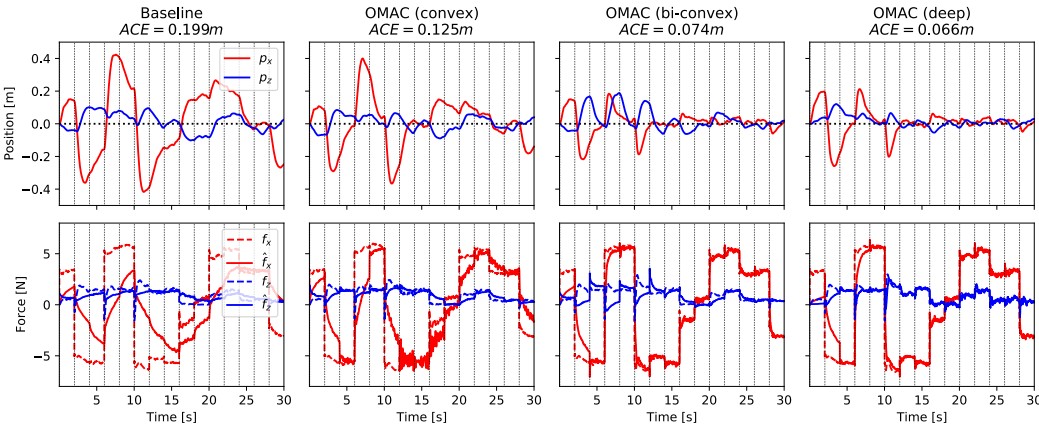

(a) xz-position trajectories (top) and force predictions (bottom) in the quadrotor experiment from one random seed. The wind condition is switched randomly every 2 s (indicated by the dotted vertical lines).

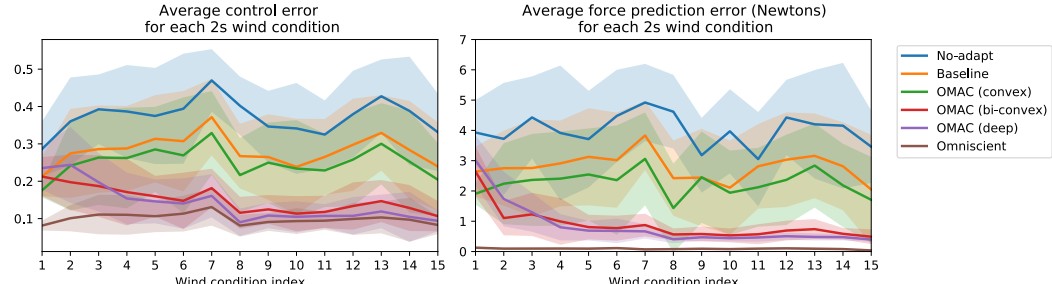

(b) The evolution of control error (left) and prediction error (right) in each wind condition. The solid lines average 10 random seeds and the shade areas show standard deviations. The performance OMAC improves as the number of wind conditions increases (especially the bi-convex and deep variants) while the baseline not.

Figure 1: Drone experiment results.

- **OMAC (deep learning)** is based on Table 4. We use a DNN with spectral normalization [38, 25, 39, 40] to represent $\phi$, and use the `Adam` optimizer [41] for $\mathcal{A}_1$. Same as other methods, $\mathcal{A}_2$ is also an OGD algorithm.

| | no-adapt | baseline | OMAC (convex) |
|---|---|---|---|
| unknown external wind | $0.663 \pm 0.142$ | $0.311 \pm 0.112$ | $0.147 \pm 0.047$ |
| | OMAC (bi-convex) | OMAC (deep) | omniscient |
| | $0.129 \pm 0.044$ | $0.093 \pm 0.027$ | $0.034 \pm 0.017$ |
| | no-adapt | baseline | OMAC (convex) |
| | $0.374 \pm 0.044$ | $0.283 \pm 0.043$ | $0.251 \pm 0.043$ |
| | OMAC (bi-convex) | OMAC (deep) | omniscient |
| | $0.150 \pm 0.019$ | $0.141 \pm 0.024$ | $0.100 \pm 0.018$ |

Table 5: ACE results in pendulum (top) and drone (bottom) experiments from 10 random seeds.

For all methods, we randomly switch the environment (wind) $c$ every 2 s. To make a fair comparison, except no-adapt or omniscient, all methods have the same learning rate for the inner-adapter $\mathcal{A}_2$ and the dimensions of $\hat{c}$ are also same ($\dim(\hat{c}) = 20$ for the pendulum and $\dim(\hat{c}) = 30$ for the drone). ACE results from 10 random seeds are depicted in Table 5. Figure 1 visualizes the drone experiment results. We observe that i) OMAC significantly outperforms the baseline; ii) OMAC adapts faster and faster as it encounters more environments but baseline cannot benefit from prior experience, especially for the bi-convex and deep versions (see Figure 1), and iii) Deep OMAC achieves the best ACE due to the representation power of DNN.

We note that in the drone experiments the performance of OMAC (convex) is only marginally better than the baseline. This is because the aerodynamic disturbance force in the quadrotor simulation is a multiplicative combination of the relative wind speed, the drone attitude, and the motor speeds; thus, the superposition structure $\hat{f}_t^{(i)} = Y_1(x_t^{(i)})\hat{\Theta}^{(i)} + Y_2(x_t^{(i)})\hat{c}_t^{(i)}$ cannot easily model the unknown force, while the bi-convex and deep learning variants both learn good controllers. In particular, OMAC (bi-convex) achieves similar performance as the deep learning case with much fewer parameters. On the other hand, in the pendulum experiments, OMAC (convex) is relatively better because a large component of the $c$-invariant part in the unknown dynamics is in superposition with the $c$-dependent part. For more details and the pendulum experiment visualization we refer to the Appendix A.7.

# 6 Related work

**Meta-learning and representation learning.** Empirically, representation learning and meta-learning have shown great success in various domains [7]. In terms of control, meta-reinforcement learning is able to solve challenging mult-task RL problems [8–10]. We remark that learning representation for control also refers to learn *state* representation from rich observations [42–44], but we consider *dynamics* representation in this paper. On the theoretic side, [4, 5, 33] have shown representation learning reduces sample complexity on new tasks, and "task diversity" is critical. Consistently, we show OMAC enjoys better convergence theoretically (Corollary 4) and empirically, and Theorem 6 also implies the importance of *environment diversity*. Another relevant line of theoretical work [45–47] uses tools from online convex optimization to show guarantees for gradient-based meta-learning, by assuming closeness of all tasks to a single fixed point in parameter space. We eliminate this assumption by considering a hierarchical or nested online optimization procedure.

**Adaptive control and online control.** There is a rich body of literature studying Lyapunov stability and asymptotic convergence in adaptive control theory [11, 12]. Recently, there has been increased interest in studying online adaptive control with non-asymptotic metrics (e.g., regret) from learning theory, largely for settings with linear systems such as online LQR or LQG with unknown dynamics [14–16, 24, 48]. The most relevant work [13] gives the first regret bound of adaptive nonlinear control with unknown nonlinear dynamics and stochastic noise. Another relevant work studies online robust control of nonlinear systems with a mistake guarantee on the number of robustness failures [49]. However, all these results focus on the single-task case. To our knowledge, we show the first non-asymptotic convergence result for multi-task adaptive control. On the empirical side, [1, 31] combine adaptive nonlinear control with meta-learning, yielding encouraging experimental results.

**Online matrix factorization.** Our work bears affinity to online matrix factorization, particularly the bandit collaborative filtering setting [30, 50, 29]. In this setting, one typically posits a linear low-rank projection as the target representation (e.g., a low-rank factorization of the user-item matrix), which is similar to our bilinear case. Setting aside the significant complexity introduced by nonlinear control, a key similarity comes from viewing different users as "tasks" and recommended items as "actions". Prior work in this area has by and large not been able to establish strong regret bounds, in part due to the non-identifiability issue inherent in matrix factorization. In contrast, in our setting, one set of latent variables (e.g., the wind condition) has a concrete physical meaning that we are allowed to measure (`ObserveEnv` in Algorithm 1), thus side-stepping this non-identifiability issue.

# 7 Concluding remarks

We have presented OMAC, a meta-algorithm for adaptive nonlinear control in a sequence of unknown environments. We provide different instantiations of OMAC under varying assumptions and conditions, leading to the first non-asymptotic convergence guarantee for multi-task adaptive nonlinear control, and integration with deep learning. We also validate OMAC empirically. We use the average control error (ACE) metric and focus on fully-actuated systems in this paper. Future work will seek to consider general cost functions and systems. It is also interesting to study end-to-end convergence guarantees of deep OMAC, with ideas from deep representation learning theories. Another interesting direction is to study how to incorporate other areas of control theory, such as robust control.

**Broader impacts.** This work is primarily focused on establishing a theoretical understanding of meta-adaptive control. Such fundamental work will not directly have broader societal impacts.

## Acknowledgments and Disclosure of Funding

This project was supported in part by funding from Raytheon and DARPA PAI, with additional support for Guanya Shi provided by the Simoudis Discovery Prize. There is no conflict of interest.

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
