# A  Appendix

## A.1  Proof of Lemma 1

*Proof.*  Using the e-ISS property in Assumption 1, we have:

$$
\begin{aligned}
\frac{1}{TN}\sum_{i=1}^{N}\sum_{t=1}^{T}\|x_t^{(i)}\| &\leq \frac{1}{TN}\sum_{i=1}^{N}\sum_{t=1}^{T}\left(\gamma\sum_{k=1}^{t-1}\rho^{t-1-k}\|B_k^{(i)}u_k^{(i)}-f_k^{(i)}+w_k^{(i)}\|\right)\\
&\overset{(a)}{\leq}\frac{\gamma}{1-\rho}\frac{1}{TN}\sum_{i=1}^{N}\sum_{t=1}^{T-1}\|B_t^{(i)}u_t^{(i)}-f_t^{(i)}+w_t^{(i)}\|\\
&\overset{(b)}{\leq}\frac{\gamma}{1-\rho}\sqrt{\frac{1}{TN}}\sqrt{\sum_{i=1}^{N}\sum_{t=1}^{T}\|B_t^{(i)}u_t^{(i)}-f_t^{(i)}+w_t^{(i)}\|^2},
\end{aligned}
\tag{8}
$$

where $(a)$ and $(b)$ are from geometric series and Cauchy-Schwarz inequality respectively. $\qquad\square$

## A.2  Proof of Lemma 2

This proof is based on the proof of Theorem 4.1 in [28].

*Proof.*  For any $\bar{\Theta}\in\mathcal{K}_1$ and $\bar{c}^{(1:N)}\in\mathcal{K}_2$ we have

$$
\begin{aligned}
&\sum_{i=1}^{N}\sum_{t=1}^{T}\ell_t^{(i)}(\hat{\Theta}^{(i)},\hat{c}_t^{(i)})-\sum_{i=1}^{N}\sum_{t=1}^{T}\ell_t^{(i)}(\bar{\Theta},\bar{c}^{(i)})\\
&\overset{(a)}{\leq}\sum_{i=1}^{N}\sum_{t=1}^{T}\nabla_{\hat{\Theta}}\ell_t^{(i)}(\hat{\Theta}^{(i)},\hat{c}_t^{(i)})\cdot(\hat{\Theta}^{(i)}-\bar{\Theta})+\sum_{i=1}^{N}\sum_{t=1}^{T}\nabla_{\hat{c}}\ell_t^{(i)}(\hat{\Theta}^{(i)},\hat{c}_t^{(i)})\cdot(\hat{c}_t^{(i)}-\bar{c}^{(i)})\\
&=\sum_{i=1}^{N}\left[G^{(i)}(\hat{\Theta}^{(i)})-G^{(i)}(\bar{\Theta})\right]+\sum_{i=1}^{N}\sum_{t=1}^{T}\left[g_t^{(i)}(\hat{c}_t^{(i)})-g_t^{(i)}(\bar{c}^{(i)})\right]\\
&\leq\underbrace{\sum_{i=1}^{N}G^{(i)}(\hat{\Theta}^{(i)})-\min_{\Theta\in\mathcal{K}_1}\sum_{i=1}^{N}G^{(i)}(\Theta)}_{\text{the total regret of }\mathcal{A}_1,\,T\cdot o(N)}+\underbrace{\sum_{i=1}^{N}\sum_{t=1}^{T}g_t^{(i)}(\hat{c}_t^{(i)})-\sum_{i=1}^{N}\min_{c^{(i)}\in\mathcal{K}_2}\sum_{t=1}^{T}g_t^{(i)}(c^{(i)})}_{\text{the total regret of }\mathcal{A}_2,\,N\cdot o(T)}.
\end{aligned}
\tag{9}
$$

where we have $(a)$ because $\ell_t^{(i)}$ is convex. Note that the total regret of $\mathcal{A}_1$ is $T\cdot o(N)$ because $G^{(i)}$ is scaled up by a factor of $T$. $\qquad\square$

## A.3  Proof of Theorem 3

*Proof.*  Since $\Theta\in\mathcal{K}_1$ and $c^{(1:N)}\in\mathcal{K}_2$, applying Lemma 2 we have

$$
\sum_{i=1}^{N}\sum_{t=1}^{T}\ell_t^{(i)}(\hat{\Theta}^{(i)},\hat{c}_t^{(i)})-\sum_{i=1}^{N}\sum_{t=1}^{T}\ell_t^{(i)}(\Theta,c^{(i)})\leq T\cdot o(N)+N\cdot o(T)
\tag{10}
$$

Recall that the definition of $\ell_t^{(i)}$ is $\ell_t^{(i)}(\hat{\Theta},\hat{c})=\|F(\phi(x_t^{(i)};\hat{\Theta}),\hat{c})-y_t^{(i)}\|^2$, and $y_t^{(i)}=f_t^{(i)}-w_t^{(i)}$. Therefore we have

$$
\begin{aligned}
\ell_t^{(i)}(\hat{\Theta}^{(i)},\hat{c}_t^{(i)})&=\|\hat{f}_t^{(i)}-f_t^{(i)}+w_t^{(i)}\|^2=\|B_t^{(i)}u_t^{(i)}-f_t^{(i)}+w_t^{(i)}\|^2\\
\ell_t^{(i)}(\Theta,c^{(i)})&=\|w_t^{(i)}\|^2\leq W^2.
\end{aligned}
\tag{11}
$$

Then applying Lemma 1 we have

$$\mathsf{ACE} \le \frac{\gamma}{1-\rho} \sqrt{\frac{\sum_{i=1}^{N} \sum_{t=1}^{T} \|B_t^{(i)} u_t^{(i)} - f_t^{(i)} + w_t^{(i)}\|^2}{TN}}$$

$$= \frac{\gamma}{1-\rho} \sqrt{\frac{\sum_{i=1}^{N} \sum_{t=1}^{T} \ell_t^{(i)}(\hat{\Theta}^{(i)}, \hat{c}_t^{(i)})}{TN}}$$

$$\overset{(a)}{\le} \frac{\gamma}{1-\rho} \sqrt{\frac{T \cdot o(N) + N \cdot o(T) + \sum_{i=1}^{N} \sum_{t=1}^{T} \ell_t^{(i)}(\Theta, c^{(i)})}{TN}}$$

$$\le \frac{\gamma}{1-\rho} \sqrt{W^2 + \frac{o(T)}{T} + \frac{o(N)}{N}}, \tag{12}$$

where $(a)$ uses (10). $\qquad\square$

## A.4 Proof of Corollary 4

Before the proof, we first present a lemma [27] which shows that the regret of an Online Gradient Descent (OGD) algorithm.

**Lemma 7** (Regret of OGD [27]). *Suppose $f_{1:T}(x)$ is a sequence of differentiable convex cost functions from $\mathbb{R}^n$ to $\mathbb{R}$, and $\mathcal{K}$ is a convex set in $\mathbb{R}^n$ with diameter $D$, i.e., $\forall x_1, x_2 \in \mathcal{K}, \|x_1 - x_2\| \le D$. We denote by $G > 0$ an upper bound on the norm of the gradients of $f_{1:T}$ over $\mathcal{K}$, i.e., $\|\nabla f_t(x)\| \le G$ for all $t \in [1, T]$ and $x \in \mathcal{K}$.*

*The OGD algorithm initializes $x_1 \in \mathcal{K}$. At time step $t$, it plays $x_t$, observes cost $f_t(x_t)$, and updates $x_{t+1}$ by $\Pi_{\mathcal{K}}(x_t - \eta_t \nabla f_t(x_t))$ where $\Pi_{\mathcal{K}}$ is the projection onto $\mathcal{K}$, i.e., $\Pi_{\mathcal{K}}(y) = \arg\min_{x \in \mathcal{K}} \|x - y\|$. OGD with learning rates $\{\eta_t = \frac{D}{G\sqrt{t}}\}$ guarantees the following:*

$$\sum_{t=1}^{T} f_t(x_t) - \min_{x^* \in \mathcal{K}} \sum_{t=1}^{T} f_t(x^*) \le \frac{3}{2} GD\sqrt{T}. \tag{13}$$

Define $\mathcal{R}(\mathcal{A}_1)$ as the total regret of the outer-adapter $\mathcal{A}_1$, and $\mathcal{R}(\mathcal{A}_2)$ as the total regret of the inner-adapter $\mathcal{A}_2$. Recall that in Theorem 3 we show that $\mathsf{ACE}(\mathsf{OMAC}) \le \frac{\gamma}{1-\rho} \sqrt{W^2 + \frac{\mathcal{R}(\mathcal{A}_1) + \mathcal{R}(\mathcal{A}_2)}{TN}}$. Now we will prove Corollary 4 by analyzing $\mathcal{R}(\mathcal{A}_1)$ and $\mathcal{R}(\mathcal{A}_2)$ respectively.

*Proof of Corollary 4.* Since the true dynamics $f(x, c^{(i)}) = Y_1(x)\Theta + Y_2(x)c^{(i)}$, we have

$$\ell_t^{(i)}(\hat{\Theta}, \hat{c}) = \|Y_1(x_t^{(i)})\hat{\Theta} + Y_2(x_t^{(i)})\hat{c} - Y_1(x_t^{(i)})\Theta - Y_2(x_t^{(i)})c^{(i)} + w_t^{(i)}\|^2. \tag{14}$$

Recall that $g_t^{(i)}(\hat{c}) = \nabla_{\hat{c}} \ell_t^{(i)}(\hat{\Theta}^{(i)}, \hat{c}_t^{(i)}) \cdot \hat{c}$, which is convex (linear) w.r.t. $\hat{c}$. The gradient of $g_t^{(i)}$ is upper bounded as

$$\|\nabla_{\hat{c}} g_t^{(i)}\| = \left\| 2Y_2(x_t^{(i)})^\top \left( Y_1(x_t^{(i)})\hat{\Theta}^{(i)} + Y_2(x_t^{(i)})\hat{c}_t^{(i)} - Y_1(x_t^{(i)})\Theta - Y_2(x_t^{(i)})c^{(i)} + w_t^{(i)} \right) \right\|$$

$$\le 2K_2 K_1 K_\Theta + 2K_2^2 K_c + 2K_2 K_1 K_\Theta + 2K_2^2 K_c + 2K_2 W$$

$$= \underbrace{4K_1 K_2 K_\Theta + 4K_2^2 K_c + 2K_2 W}_{C_2}. \tag{15}$$

From Lemma 7, using learning rates $\eta_t^{(i)} = \frac{2K_c}{C_2 \sqrt{t}}$ for all $i$, the regret of $\mathcal{A}_2$ at each outer iteration is upper bounded by $3K_c C_2 \sqrt{T}$. Then the total regret of $\mathcal{A}_2$ is bounded as

$$\mathcal{R}(\mathcal{A}_2) \le 3K_c C_2 N\sqrt{T}. \tag{16}$$

Now let us study $\mathcal{A}_1$. Similarly, recall that $G^{(i)}(\hat{\Theta}) = \sum_{t=1}^{T} \nabla_{\hat{\Theta}} \ell_t^{(i)}(\hat{\Theta}^{(i)}, \hat{c}_t^{(i)}) \cdot \hat{\Theta}$, which is convex (linear) w.r.t. $\hat{\Theta}$. The gradient of $G^{(i)}$ is upper bounded as

$$
\begin{aligned}
\|\nabla_{\hat{\Theta}} G^{(i)}\| &= \left\| \sum_{t=1}^{T} 2Y_1(x_t^{(i)})^\top \left( Y_1(x_t^{(i)})\hat{\Theta}^{(i)} + Y_2(x_t^{(i)})\hat{c}_t^{(i)} - Y_1(x_t^{(i)})\Theta - Y_2(x_t^{(i)})c^{(i)} + w_t^{(i)} \right) \right\| \\
&\leq T \left( 2K_1^2 K_\Theta + 2K_1 K_2 K_c + 2K_1^2 K_\Theta + 2K_1 K_2 K_c + 2K_1 W \right) \\
&= T \left( \underbrace{4K_1^2 K_\Theta + 4K_1 K_2 K_c + 2K_1 W}_{C_1} \right).
\end{aligned}
$$

(17)

From Lemma 7, using learning rates $\bar{\eta}^{(i)} = \frac{2K_\Theta}{TC_1\sqrt{i}}$, the total regret of $\mathcal{A}_1$ is upper bounded as

$$
\mathcal{R}(\mathcal{A}_1) \leq 3K_\Theta T C_1 \sqrt{N}.
$$

(18)

Finally using Theorem 3 we have

$$
\begin{aligned}
\mathsf{ACE(OMAC)} &\leq \frac{\gamma}{1-\rho} \sqrt{W^2 + \frac{\mathcal{R}(\mathcal{A}_1) + \mathcal{R}(\mathcal{A}_2)}{TN}} \\
&\leq \frac{\gamma}{1-\rho} \sqrt{W^2 + 3(K_\Theta C_1 \frac{1}{\sqrt{N}} + K_c C_2 \frac{1}{\sqrt{T}})}.
\end{aligned}
$$

(19)

Now let us analyze $\mathsf{ACE}$(baseline adaptive control). To simplify notations, we define $\bar{Y}(x) = [Y_1(x) \ Y_2(x)] : \mathbb{R}^n \to \mathbb{R}^{n \times (p+h)}$ and $\hat{\alpha} = [\hat{\Theta}; \hat{c}] \in \mathbb{R}^{p+h}$. The baseline adaptive controller updates the whole vector $\hat{\alpha}$ at every time step. We denote the ground truth parameter by $\alpha^{(i)} = [\Theta; c^{(i)}]$, and the estimation by $\hat{\alpha}_t^{(i)} = [\hat{\Theta}_t^{(i)}; \hat{c}_t^{(i)}]$. We have $\|\alpha^{(i)}\| \leq \sqrt{K_\Theta^2 + K_c^2}$. Define $\bar{\mathcal{K}} = \{\hat{\alpha} = [\hat{\Theta}; \hat{c}] : \|\hat{\Theta}\| \leq \mathcal{K}_\Theta, \|\hat{c}\| \leq \mathcal{K}_c\}$, which is a convex set in $\mathbb{R}^{p+h}$.

Note that the loss function for the baseline adaptive control is $\bar{\ell}_t^{(i)}(\hat{\alpha}) = \|\bar{Y}(x_t^{(i)})\hat{\alpha} - Y_1(x_t^{(i)})\Theta - Y_2(x_t^{(i)})c^{(i)} + w_t^{(i)}\|^2$. The gradient of $\bar{\ell}_t^{(i)}$ is

$$
\nabla_{\hat{\alpha}} \bar{\ell}_t^{(i)}(\hat{\alpha}) = 2 \begin{bmatrix} Y_1(x_t^{(i)})^\top \\ Y_2(x_t^{(i)})^\top \end{bmatrix} (Y_1(x_t^{(i)})\hat{\Theta} + Y_2(x_t^{(i)})\hat{c} - Y_1(x_t^{(i)})\Theta - Y_2(x_t^{(i)})c^{(i)} + w_t^{(i)}),
$$

(20)

whose norm on $\bar{\mathcal{K}}$ is bounded by

$$
\sqrt{4(K_1^2 + K_2^2)(2K_1 K_\Theta + 2K_2 K_c + W)^2} = \sqrt{C_1^2 + C_2^2}.
$$

(21)

Therefore, from Lemma 7, running OGD on $\bar{\mathcal{K}}$ with learning rates $\frac{2\sqrt{K_\Theta^2 + K_c^2}}{\sqrt{C_1^2 + C_2^2}\sqrt{t}}$ gives the following guarantee at each outer iteration:

$$
\sum_{t=1}^{T} \bar{\ell}_t^{(i)}(\hat{\alpha}_t^{(i)}) - \bar{\ell}_t^{(i)}(\alpha^{(i)}) \leq 3\sqrt{K_\Theta^2 + K_c^2}\sqrt{C_1^2 + C_2^2}\sqrt{T}.
$$

(22)

Finally, similar as (12) we have

$$
\begin{aligned}
\mathsf{ACE}\text{(baseline adaptive control)} &\leq \frac{\gamma}{1-\rho} \sqrt{\frac{\sum_{i=1}^{N} \sum_{t=1}^{T} \bar{\ell}_t^{(i)}(\hat{\alpha}_t^{(i)})}{TN}} \\
&\leq \frac{\gamma}{1-\rho} \sqrt{\frac{\sum_{i=1}^{N} 3\sqrt{K_\Theta^2 + K_c^2}\sqrt{C_1^2 + C_2^2}\sqrt{T} + \sum_{i=1}^{N} \sum_{t=1}^{T} \bar{\ell}_t^{(i)}(\alpha^{(i)})}{TN}} \\
&\leq \frac{\gamma}{1-\rho} \sqrt{W^2 + 3\sqrt{K_\Theta^2 + K_c^2}\sqrt{C_1^2 + C_2^2}\frac{1}{\sqrt{T}}}.
\end{aligned}
$$

(23)

Note that this bound does not improve as the number of environments (i.e., $N$) increases. $\qquad \square$

## A.5 Proof of Theorem 5

*Proof.* For any $\Theta \in \mathcal{K}_1$ and $c^{(1:N)} \in \mathcal{K}_2$ we have

$$
\begin{aligned}
&\sum_{i=1}^{N}\sum_{t=1}^{T} \ell_t^{(i)}(\hat{\Theta}^{(i)}, \hat{c}_t^{(i)}) - \sum_{i=1}^{N}\sum_{t=1}^{T} \ell_t^{(i)}(\Theta, c^{(i)}) \\
&= \sum_{i=1}^{N}\sum_{t=1}^{T} \left[ \ell_t^{(i)}(\hat{\Theta}^{(i)}, \hat{c}_t^{(i)}) - \ell_t^{(i)}(\hat{\Theta}^{(i)}, c^{(i)}) \right] + \sum_{i=1}^{N}\sum_{t=1}^{T} \left[ \ell_t^{(i)}(\hat{\Theta}^{(i)}, c^{(i)}) - \ell_t^{(i)}(\Theta, c^{(i)}) \right] \\
&= \underbrace{\sum_{i=1}^{N}\sum_{t=1}^{T} \left[ g_t^{(i)}(\hat{c}_t^{(i)}) - g_t^{(i)}(c^{(i)}) \right]}_{\leq o(T)} + \underbrace{\sum_{i=1}^{N} \left[ G^{(i)}(\hat{\Theta}^{(i)}) - G^{(i)}(\Theta) \right]}_{\leq T \cdot o(N)}
\end{aligned}
\tag{24}
$$

Then combining with Lemma 1 results in the ACE bound. $\qquad\square$

## A.6 Proof of Theorem 6

*Proof.* Note that in this case the available measurement of $f$ at the end of the outer iteration $i$ is:

$$
y_t^{(j)} = Y(x_t^{(j)})\Theta c^{(j)} - w_t^{(j)}, \quad 1 \leq j \leq i, 1 \leq t \leq T.
\tag{25}
$$

Recall that the Ridge-regression estimation of $\hat{\Theta}$ is given by

$$
\begin{aligned}
\hat{\Theta}^{(i+1)} &= \arg\min_{\hat{\Theta}} \lambda \|\hat{\Theta}\|_F^2 + \sum_{j=1}^{i}\sum_{t=1}^{T} \|Y(x_t^{(j)})\hat{\Theta}c^{(j)} - y_t^{(j)}\|^2 \\
&= \arg\min_{\hat{\Theta}} \lambda \|\hat{\Theta}\|_F^2 + \sum_{j=1}^{i}\sum_{t=1}^{T} \|Z_t^{(j)}\mathrm{vec}(\hat{\Theta}) - y_t^{(j)}\|^2.
\end{aligned}
\tag{26}
$$

Note that $y_t^{(j)} = (c^{(j)\top} \otimes Y(x_t^{(j)})) \cdot \mathrm{vec}(\Theta) - w_t^{(j)} = Z_t^{(j)}\mathrm{vec}(\Theta) - w_t^{(j)}$. Define $V_i = \lambda I + \sum_{j=1}^{i}\sum_{t=1}^{T} Z_t^{(j)\top} Z_t^{(j)}$. Then from the Theorem 2 of [32] we have

$$
\|\mathrm{vec}(\hat{\Theta}^{(i+1)} - \Theta)\|_{V_i} \leq R\sqrt{\bar{p}h \log\left(\frac{1 + iT \cdot nK_Y^2 K_c^2/\lambda}{\delta}\right)} + \sqrt{\lambda}K_\Theta
\tag{27}
$$

for all $i$ with probability at least $1 - \delta$. Note that the environment diversity condition implies: $V_i \succ \Omega(i)I$. Finally we have

$$
\|\hat{\Theta}^{(i+1)} - \Theta\|_F^2 = \|\mathrm{vec}(\hat{\Theta}^{(i+1)} - \Theta)\|^2 \leq O\left(\frac{1}{i}\right)O(\log(iT/\delta)) = O\left(\frac{\log(iT/\delta)}{i}\right).
\tag{28}
$$

Then with a fixed $\hat{\Theta}^{(i+1)}$, in outer iteration $i + 1$ we have

$$
g_t^{(i+1)}(\hat{c}) = \|Y(x_t^{(i+1)})\hat{\Theta}^{(i+1)}\hat{c} - Y(x_t^{(i+1)})\Theta c^{(i+1)} + w_t^{(i+1)}\|^2.
\tag{29}
$$

Since $\mathcal{A}_2$ gives sublinear regret, we have

$$
\begin{aligned}
&\sum_{t=1}^{T} \|Y(x_t^{(i+1)})\hat{\Theta}^{(i+1)}\hat{c}_t^{(i+1)} - Y(x_t^{(i+1)})\Theta c^{(i+1)} + w_t^{(i+1)}\|^2 \\
&- \min_{\hat{c}\in\mathcal{K}_2} \sum_{t=1}^{T} \|Y(x_t^{(i+1)})\hat{\Theta}^{(i+1)}\hat{c} - Y(x_t^{(i+1)})\Theta c^{(i+1)} + w_t^{(i+1)}\|^2 = o(T).
\end{aligned}
\tag{30}
$$

Note that

$$
\begin{aligned}
&\min_{\hat{c}\in\mathcal{K}_2} \sum_{t=1}^{T} \|Y(x_t^{(i+1)})\hat{\Theta}^{(i+1)}\hat{c} - Y(x_t^{(i+1)})\Theta c^{(i+1)} + w_t^{(i+1)}\|^2 \\
&\leq \sum_{t=1}^{T} \|Y(x_t^{(i+1)})\hat{\Theta}^{(i+1)}c^{(i+1)} - Y(x_t^{(i+1)})\Theta c^{(i+1)} + w_t^{(i+1)}\|^2 \\
&\overset{(a)}{\leq} TW^2 + T \cdot K_Y^2 \cdot O\left(\frac{\log(iT/\delta)}{i}\right) \cdot K_c^2,
\end{aligned}
\tag{31}
$$

where $(a)$ uses (28).

Finally we have

$$\sum_{t=1}^{T} \|\hat{f}_t^{(i+1)} - f_t^{(i+1)} + w_t^{(i+1)}\|^2$$

$$= \sum_{t=1}^{T} \|Y(x_t^{(i+1)})\hat{\Theta}^{(i+1)}\hat{c}_t^{(i+1)} - Y(x_t^{(i+1)})\Theta c^{(i+1)} + w_t^{(i+1)}\|^2 \tag{32}$$

$$\overset{(b)}{\leq} o(T) + TW^2 + O(T\frac{\log(iT/\delta)}{i})$$

for all $i$ with probability at least $1 - \delta$. $(b)$ is from (30) and (31). Then with Lemma 1 we have (with probability at least $1 - \delta$)

$$\mathsf{ACE} \leq \frac{\gamma}{1-\rho}\sqrt{\frac{\sum_{i=1}^{N} o(T) + TW^2 + O(T\frac{\log(iT/\delta)}{i})}{TN}}$$

$$\leq \frac{\gamma}{1-\rho}\sqrt{W^2 + \frac{o(T)}{T} + \frac{O(\log(NT/\delta))}{N}\sum_{i=1}^{N}\frac{1}{i}} \tag{33}$$

$$\leq \frac{\gamma}{1-\rho}\sqrt{W^2 + \frac{o(T)}{T} + O(\frac{\log(NT/\delta)\log(N)}{N})}.$$

If we relax the environment diversity condition to $\Omega(\sqrt{i})$, in (28) we will have $O(\frac{\log(iT/\delta)}{\sqrt{i}})$. Therefore in (33) the last term becomes $\frac{O(\log(NT/\delta))}{N}\sum_{i=1}^{N}\frac{1}{\sqrt{i}} \leq \frac{O(\log(NT/\delta))}{\sqrt{N}}$. □

## A.7 Experimental details

### A.7.1 Theoretical justification of Deep OMAC

Recall that in Deep OMAC (Table 4 in Section 5) the model class is $F(\phi(x;\hat{\Theta}),\hat{c}) = \phi(x;\hat{\Theta}) \cdot \hat{c}$, where $\phi : \mathbb{R}^n \to \mathbb{R}^{n \times h}$ is a neural network parameterized by $\hat{\Theta}$. We provide the following proposition to justify such choice of model class.

**Proposition 1.** *Let* $\bar{f}(x,\bar{c}) : [-1,1]^n \times [-1,1]^{\bar{h}} \to \mathbb{R}$ *be an analytic function of* $[x,\bar{c}] \in [-1,1]^{n+\bar{h}}$ *for* $n, \bar{h} \geq 1$. *Then for any* $\epsilon > 0$, *there exist* $h(\epsilon) \in \mathbb{Z}^+$, *a polynomial* $\bar{\phi}(x) : [-1,1]^n \to \mathbb{R}^{h(\epsilon)}$ *and another polynomial* $c(\bar{c}) : [-1,1]^{\bar{h}} \to \mathbb{R}^{h(\epsilon)}$ *such that*

$$\max_{[x,\bar{c}]\in[-1,1]^{n+\bar{h}}} \|\bar{f}(x,\bar{c}) - \bar{\phi}(x)^{\top}c(\bar{c})\| \leq \epsilon$$

*and* $h(\epsilon) = O((\log(1/\epsilon))^{\bar{h}})$.

Note that here the dimension of $c$ depends on the precision $1/\epsilon$. In practice, for OMAC algorithms, the dimension of $\hat{c}$ or $c$ (i.e., the latent space dimension) is a hyperparameter, and not necessarily equal to the dimension of $\bar{c}$ (i.e., the dimension of the actual environmental condition). A variant of this proposition is proved in [34]. Since neural networks are universal approximators for polynomials, this theorem implies that the structure $\phi(x;\hat{\Theta})\hat{c}$ can approximate any analytic function $\bar{f}(x,\bar{c})$, and the dimension of $\hat{c}$ only increases polylogarithmically as the precision increases.

### A.7.2 Pendulum dynamics model and controller design

In experiments, we consider a nonlinear pendulum dynamics with unknown gravity, damping and external 2D wind $w = [w_x; w_y] \in \mathbb{R}^2$. The continuous-time dynamics model is given by

$$ml^2\ddot{\theta} - ml\hat{g}\sin\theta = u + \underbrace{f(\theta, \dot{\theta}, c(w))}_{\text{unknown}}, \tag{34}$$

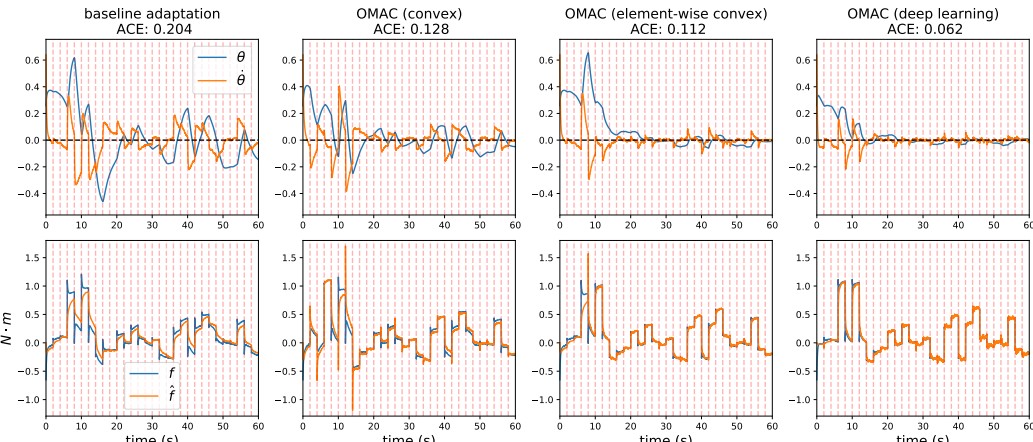

Figure 2: Trajectories (top) and force predictions (bottom) in the pendulum experiment from one random seed. The wind condition is switched randomly every 2 s (indicated by the dashed red lines). The performance of OMAC improves as it encounters more environments while baseline not.

where

$$f(\theta, \dot{\theta}, c(w)) = \underbrace{\vec{l} \times F_{\text{wind}}}_{\text{air drag}} - \underbrace{\alpha_1 \dot{\theta}}_{\text{damping}} + \underbrace{ml(g - \hat{g}) \sin \theta}_{\text{gravity mismatch}},$$

$$F_{\text{wind}} = \alpha_2 \cdot \|r\|_2 \cdot r, r = w - \begin{bmatrix} l\dot{\theta} \cos \theta \\ -l\dot{\theta} \sin \theta \end{bmatrix}. \tag{35}$$

This model generalizes the pendulum with external wind model in [35] by introducing extra modelling mismatches (e.g., gravity mismatch and unknown damping). In this model, $\alpha_1$ is the damping coefficient, $\alpha_2$ is the air drag coefficient, $r$ is the relative velocity of the pendulum to the wind, $F_{\text{wind}}$ is the air drag force vector, and $\vec{l}$ is the pendulum vector. Define the state of the pendulum as $x = [\theta; \dot{\theta}]$. The discrete dynamics is given by

$$x_{t+1} = \begin{bmatrix} \theta_t + \delta \cdot \dot{\theta}_t \\ \dot{\theta}_t + \delta \cdot \frac{ml\hat{g} \sin \theta_t + u_t + f(\theta_t, \dot{\theta}_t, c)}{ml^2} \end{bmatrix} = \underbrace{\begin{bmatrix} 1 & \delta \\ 0 & 1 \end{bmatrix}}_{A} x_t + \underbrace{\begin{bmatrix} 0 \\ \frac{\delta}{ml^2} \end{bmatrix}}_{B} (u_t + ml\hat{g} \sin \theta_t + f(x_t, c)), \tag{36}$$

where $\delta$ is the discretization step. We use the controller structure $u_t = -Kx_t - ml\hat{g} \sin \theta_t - \hat{f}$ for all 6 controllers in the experiments, but different controllers have different methods to calculate $\hat{f}$ (e.g., the **no-adapt** controller uses $\hat{f} = 0$ and the **omniscient** one uses $\hat{f} = f$). We choose $K$ such that $A - BK$ is stable (i.e., the spectral radius of $A - BK$ is strictly smaller than 1), and then the e-ISS assumption in Assumption 1 naturally holds. We visualize the pendulum experiment results in fig. 2.

### A.7.3 Quadrotor dynamics model and controller design

Now we introduce the quadrotor dynamics with aerodynamic disturbance. Consider states given by global position, $p \in \mathbb{R}^3$, velocity $v \in \mathbb{R}^3$, attitude rotation matrix $R \in \text{SO}(3)$, and body angular velocity $\omega \in \mathbb{R}^3$. Then dynamics of a quadrotor are

$$\dot{p} = v, \qquad\qquad m\dot{v} = mg + Rf_T + f, \tag{37a}$$

$$\dot{R} = RS(\omega), \qquad\qquad J\dot{\omega} = J\omega \times \omega + \tau, \tag{37b}$$

where $m$ is the mass, $J$ is the inertia matrix of the quadrotor, $S(\cdot)$ is the skew-symmetric mapping, $g$ is the gravity vector, $f_T = [0, 0, T]^\top$ and $\tau = [\tau_x, \tau_y, \tau_z]^\top$ are the total thrust and body torques from four rotors, and $f = [f_x, f_y, f_z]^\top$ are forces resulting from unmodelled aerodynamic effects and varying wind conditions. In the simulator, $f$ is implemented as the aerodynamic model given in [36].

**Controller design.** Quadrotor control, as part of multicopter control, generally has a cascaded structure to separate the design of the position controller, attitude controller, and thrust mixer

(allocation). In this paper, we incorporate the online learned aerodynamic force $\hat{f}$ in the position controller via the following equation:

$$f_d = -mg - m(K_P \cdot p + K_D \cdot v) - \hat{f}, \tag{38}$$

where $K_P, K_D \in \mathbb{R}^{3\times3}$ are gain matrices for the PD nominal term, and different controllers have different methods to calculate $\hat{f}$ (e.g., the **omniscient** controller uses $\hat{f} = f$). Given the desired force $f_d$, a kinematic module decomposes it into the desired $R_d$ and the desired thrust $T_d$ so that $R_d \cdot [0, 0, T_d]^\top \approx f_d$. Then the desired attitude and thrust are sent to a lower level attitude controller (e.g., the attitude controller in [51]).