# OpenReview forum: "Meta-Adaptive Nonlinear Control: Theory and Algorithms"
_NeurIPS.cc/2021/Conference — NeurIPS 2021 Poster_

### Official Review · Reviewer_H6i6 · 2021-07-01

**Rating:** 7
**Confidence:** 4

**Summary:**

The paper proposes a novel algorithm for adaptive nonlinear control that can learn to stabilize a sequence of dynamical systems that share a common part of their dynamics, but differ in another part that can be estimated from data. It is assumed that the unknown part can be learned in terms of a set of shared features, which also need to be learned from data. The algorithm is decomposed into an outer and inner loop, where features are learned in the outer loop, and the specific component of the dynamics for a given environment is learned in an inner loop. The authors consider several approximation schemes for the unknown dynamical component (convex, element-wise convex, and a deep neural net), and prove asymptotic convergence for each of them. Empirical verification on the problem of stabilizing a pendulum with unknown wind acting on it demonstrates that the richest function approximation, the deep neural net, achieves the lowest control error, which is only greater than that of an omniscient controller by a factor of about 3.

**Ethical Concerns:**

I do not see any particular ethical concerns with this paper, beyond general concerns about potential dual use of control technology and robotics systems. This is mainly theoretical work that is far removed from potential dangerous applications.

**Limitations And Societal Impact:**

The authors have made a good effort to assess the limitations and potential negative societal impact of their work.

**Main Review:**

The main contribution of the paper is probably the convergence guarantee of the proposed learning algorithm. This appears to be the first in the field. The approximation schemes for the unknown parts of the system dynamics are well known from previous research. Related work is adequately cited, and connections are made to both work in the field of adaptive control, as well as representation learning in the fields of machine and reinforcement learning. In fact, another merit of this paper is that it draws on research from both fields to propose a unified view on the problem of meta-learning of system dynamics.

The submission appears to be technically sound, and the empirical verification is informative. One additional issue that the authors might address is how good these approximation schemes are with respect to the functions that they are trying to approximate. My understanding is that the results in Table 5 are averaged over all trials (as per Definition 1), which includes the learning period. How about learning all functions  $\hat{f}_t^{(i)}$ from a sufficiently large amount of data in a supervised learning manner, and then using them for control, similar to the omniscient controller? To use Example 1, for the convex case this would mean finding the parameters in Eq. 4 using supervised learning (regression, in this case). If the environment-specific functions are estimated in this way, the bias in comparison due to the varying quality of the approximation bases could be removed.

The paper is written very clearly, and is very well organized. A minor typo: line 301, "roboust" -> "robust". I believe the results are very significant, because, first, the paper provides convergence proofs, and second, the method does look practical for solving problems involving control of a family of related systems that differ by an unknown part of their dynamics. I think it is a solid advance of the state of the art in adaptive control and meta learning.

Update: I am satisfied with the effort the authors made to investigate the effect of learning the features in a supervised learning manner, and I think they have addressed my review adequately. Seeing that my rating was a bit of an outlier in comparison with the others, I have revised it to be more in line with them.

**Time Spent Reviewing:**

2

---

> ### Author Response · Authors · 2021-08-11
> **Response to Reviewer H6i6**
>
> We would like to thank the reviewer for the positive comments on the clarity, organization, and novelty of this paper. We agree with the reviewer that the high-level goal of this paper is to build a novel unified view on meta-learning/representation learning and adaptive control.
> Based on the reviewer's suggestion, we provide a detailed study on how well the proposed approximation models learn the residual dynamics $f(x,c^{(i)})$ in a supervised learning manner. We will incorporate the results into the paper. We believe this further study strengthens the presentation and results in the paper.

---

> > ### Comment · Reviewer_H6i6 · 2021-08-18
> > **Thank you for addressing my suggestion**
> >
> > Thank you for investigating the effect of learning the features in a supervised learning manner, I think that it does indeed strengthen the paper.

---

### Official Review · Reviewer_UjXF · 2021-07-16

**Rating:** 7
**Confidence:** 4

**Summary:**

This paper proposes and analyzes a multi-task learning approach to adaptive nonlinear control under changing environments.  The paper assumes that the dynamics can be decomposed as the sum of a known (exponentially input-to-state-stable) component and an unknown and changing environment dependent component.  The adaptive control problem is conducted over N episodes (the outer loop) of T time steps (the inner loop).  By assuming that the unknown environmental dependent component can be defined in terms of a set of common features across environments $\phi$ and an environment dependent term $c^{i}$ for the i-th environment, a modular interaction protocol, called Online Meta-Adaptive Control (OMAC) is suggested wherein the outer loop is treated as an online learning problem with the goal of learning the shared representation $\phi$, and the inner loop is viewed as an adaptive control problem wherein the environment dependent component $c^{i}$ is the unknown term to be learned.  It is then shown under suitable convexity assumptions that the Average Control Error (the average of the state norms over both inner and outer iterations) achieved by OMAC is sublinear in both the number episodes $N$ and the task horizon length $T$.  A deep learning based approach for learning the common features $\phi$ is also proposed.  The paper concludes with an illustrative numerical example (an uncertain inverted pendulum) comparing various flavors of OMAC to non-adaptive and traditional adaptive control methods.

**Limitations And Societal Impact:**

See above regarding limitations with respect to translating theory into practice.

**Main Review:**


I thoroughly enjoyed reading this paper.  I found it to be clearly written, and the problem being posed and addressed to be of significant interest to the learning and control community.  The results are rigorously stated and technically correct.  In my view, the main novelty/contributions of the paper are (i) defining a formalism for reasoning about meta-learning in the context of nonlinear adaptive controls and (b) recognizing that once suitably posed, the resulting problem can be naturally addressed using (nested) online convex optimization techniques, leading to clean and intuitive theorem statements and proofs.

That being said, I do have some questions/comments that I would like to have clarified (and I believe would also help improve the quality of the paper if addressed):

- How would the results change for a system that isn’t fully actuated?  This is a very strong assumption that not even the simple inverted pendulum example used in the numerical experiments section satisfied.

- The assumption of exponential input-to-state-stability (E-ISS) is not standard in nonlinear adaptive control: it is in fact a very strong assumption made in [12] (Boffi et al.) that may also be difficult satisfy.  It is also not appropriate to cite [12] as a “standard nonlinear adaptive control” reference, as it is very recent and makes many non-standard assumptions, including E-ISS.  I want to emphasize that I have no problem with this assumption being made, especially when attempting to derive a first result for a novel problem formulation, but take issue with its framing as being standard.

- A comment is made early on that initial conditions can be assumed to be 0 wlog, as by E-ISS, they only result in a constant additive term.  This is obviously true if all initial conditions are drawn from a compact set, e.g., that $||x^{i}_0||\leq B_0$ for all episodes $i$ and some $B_0>0$.  However, the “environment switching” process suggested in the experimental section is that episodes do not require hard resets (specifically, a 60s task is divided into 30 x 2s episodes) — it is therefore not clear that the assumption of $||x^{i}_0||\leq B_0$ holds anymore.  In particular, if $f_0 + f - \hat{f}$ is unstable, then the state may be blowing up on you, in which case $||x^{i}_0||\leq B_0$ is no longer obviously true.

- Related to the above question, if hard resets indeed end up being required to ensure the desired error bounds, then is it realistic to ask for $N>T$?  For a 10s task with sampling time of 0.1s, this would mean a minimum of 100 novel environments.  Similarly, what was the sampling time $\delta$ used to discretize the continuous time dynamics for the inverted pendulum used in Section 5: was it such that $30>2/\delta$?

- It would be good to see at least an informal quantification of the computational complexity tradeoffs associated with choosing larger or smaller $N$ vs. $T$.  In particular, if I consider the example in Section 5 conducted over an overall horizon of $H=60/\delta$, why shouldn’t I set $N=60$ and $T=1/\delta$ instead of $N=30$ and $T=2/\delta$?  Given that the model for environmental dependent dynamics assumes that the environment specific term $c$ is constant over an episode, this suggests smaller $T$ and larger $N$ will lead to both better bounds relative to standard adaptive control, and a more dynamic controller.  However, unless I’m misunderstanding, we recover standard adaptive control in the setting where $N=H$ and $T=1$.  I believe what’s missing from the theory is capturing the computational complexity of the approach (which increases with $N$ as this means more meta-updates over an overall horizon $H$), but I’m not sure.



**Time Spent Reviewing:**

5

---

> ### Author Response · Authors · 2021-08-11
> **Response to Reviewer UjXF**
>
> Thank you for the review and comments. We answer the reviewer’s questions/comments separately as follows (in the same order as the reviewer’s questions):
>
> * Beyond Eq. (1), the OMAC framework can be used for general robotics systems with matched uncertainty, where $B$ is not necessarily full rank, in the form $$x_{t+1}=f_0(x_t)+B_t(u_t-f(x_t,c))+w_t.$$ We will add formal justifications for a variety of under-actuated examples in the paper, such as pendulums and drones.
> To summarize, our model supports not only general fully-actuated systems in Eq. (1), but the above dynamics with matched uncertainty. Meta-adaptive control for other under-actuated systems requires something beyond direct disturbance compensation (i.e., it needs some multi-step planning), and is definitely an interesting future direction. We will add a remark in the paper.
>
> * Thank you for pointing these out. We will rephrase the remark in lines 92-98, and state that such an assumption is in fact strong (in nonlinear systems). We will address the reviewer’s concern about citations.
>
> * As the reviewer mentioned, in the experiment we don’t reset the initial state. Since $f_0$ is E-ISS and $f-\hat{f}$ is bounded, therefore, $x_0^{(i)}$ is bounded. The boundedness of $f$ and $\hat{f}$ indeed is incorporated in all the theoretical statements. We will elaborate on this in lines 90-91.
>
> * As we mentioned in the above bullet point, since $f_0$ is E-ISS and $f-\hat{f}$ is bounded, the case $N>T$ is valid. Regarding the discretization step, we use the common (in robotics) choice of $\delta=0.01$.
>
> * In the OMAC framework, the inner iteration length $T$ is pre-specified by the domain. For example, in our experiment setting, $T=2$,  i.e., the wind switches every 2 seconds, and the learning algorithm does not have any control on what $T$ is. Experiment designer sets $T$ in advance.
> We are definitely interested in the setting the reviewer proposed, i.e., the wind continuously changes every time step (e.g., a Wiener process), and $T$ is rather a hyperparameter. We will add an ablation analysis in the supplementary material to show the trade-off between adaptability and computations as $T$ decreases.

---

### Official Review · Reviewer_ZMnJ · 2021-07-17

**Rating:** 6
**Confidence:** 3

**Summary:**

This paper presents an approach to adaptive nonlinear control in a changing environment: an agent must regulate a nonlinear dynamical system which changes from episode to episode. The work focuses on fully actuated, control affine systems, and takes an approach which aims to identify the true dynamics of the system and choose controls to cancel out the effect of these dynamics to regulate the system. The approach involves multi-task online learning, whereby non-varying elements of the dynamics are learned between episodes, while during an episode, the model adapts to estimate the part of the dynamics that vary.

The authors show that depending on the structure of the unknown dynamics, tools from online convex optimization can yield algorithms with sublinear cumulative control error bounds. The same structure can be used with a neural network model, albeit without guarantees on performance.

**Limitations And Societal Impact:**

The discussion of limitations could be expanded to highlight the strong assumption made in the theoretical results that the true system is in the modeling class. Given the theoretical nature of this work, the lack of a section on negative societal impact is alright.

**Main Review:**

The approach takes a clean formulation to the problem of adaptive control in environments that change from episode to episode. The choice of control scheme (certainty equivalent) and focus on fully actuated control affine systems allow for a straightforward application of results from the online convex optimization literature to this domain. I thought the organization of the paper was good, starting from the general formulation, and clearly stating the assumptions for each instantiation before detailing the theoretical results. While making strong assumptions (that the true system lies in the model class without any model mismatch), there are some interesting theoretical insights, such as the connection of environment diversity to cumulative regret, and the results guiding the selection of learning rates for Corollary 4.

However, the paper does have some weaknesses, primarily related to the connection of the theoretical results to practice:
- First, the experimental results do not reflect the theoretical procedure except in the joint convex case. The assumption that for an element-wise convex system that the true value of c would be presented after the trial is strong, and theoretical results on handling the error from using $\hat{c}^{(T)}$ would strengthen the paper.
- Corollary 4 considers an adaptive control baseline which optimizes both Theta and c. However, in the experiments the Baseline only adapts c, and not both. This severely limits the representational capacity of the Baseline, and not a fair comparison. Indeed, the impressive performance of the deep learning solution suggest that representational capacity is key to performing well on the chosen experiment.
- The experimental domain is very simple with only 2 states, and a nominal model that is very accurate, apart from minor mismatch in wind and gravity. Furthermore, it does not match the theory, as it is under actuated (B is not full rank). It would have been helpful to see how the algorithm scales to larger state and action spaces.
- Comparison to other meta-adaptive control literature is missing. Even though many of these works do not perform online learning, it would be helpful to see the difference in ACE in running the nominal controller for K episodes to gather data, meta-training, and subsequently deploying the meta-trained adaptive controller.

Overall, this paper presents what an application of results from the OCO literature to meta-adaptive nonlinear control. While this connection is novel to me, the paper is hindered by limited experiments which do not showcase the applicability of the theoretical results to realistic systems, and do not compare to strong enough baselines.

**Needs Ethics Review:**

Yes

**Time Spent Reviewing:**

3

---

> ### Author Response · Authors · 2021-08-10
> **Response to Reviewer ZMnJ**
>
> Thank you for the review and comments. We answer the reviewer’s questions separately as follows (in the same order as the reviewer’s questions):
>
> * Regarding the comment about analyzing the error form using $\hat{c}^{(i)}$ instead of $c^{(i)}$, the non-identifiability issues in the bi-convex case makes it very challenging (see line 198-203 and paper [5]), because the “optimal” $\hat{c}^{(i)}$ is not identifiable and not unique. Note that in experiments we use $\hat{c}^{(i)}$ for the bi-convex setting to make a fair comparison, since the convex and deep learning settings use $\hat{c}^{(i)}$.
> Moreover, as discussed in line 285-292, in robotics, the environment condition $c^{(i)}$ often has a concrete physical meaning (e.g., the wind condition in [1] and [27]), and we only need to measure $c^{(i)}$ after the outer iteration $(i)$ is finished, which has a lot of applications (e.g., [1,2] and [27]). Therefore, while the assumption of observing $c^{(i)}$ is strong, it holds for many real-world applications.
>
> * In order to follow the suggestion, we conducted the mentioned empirical study where both $\hat{\Theta}$ and $\hat{c}$ are updated at each step in the convex setting (i.e., the baseline adaptive control in Corollary 4). Note that such baseline adaptive control requires more computations than ours. We observed that the OMAC framework provides superior performance which is consistent with the claim in Corollary 4. We will incorporate the results in the paper. The current baseline in Table 5 is a special case of OMAC in the bi-convex case, thereby showing the importance of the meta-adapter.
> Moreover, as suggested by reviewer H6i6, we will add another ablation analysis by showing the offline batch learning results of all methods (i.e., collecting $K$ outer iterations’ data and do offline supervised meta-learning), and the expected result is that all methods have similar training errors. Such analysis will remove the bias from the representational capacity.
>
> * We believe the model mismatch is not minor (please refer to the performance of the “no-adaptation” method, it is much worse than other methods).
> Beyond Eq. (1), the OMAC framework can handle general robotics systems with matched uncertainty, where $B$ is not necessarily full rank, in the form $$x_{t+1}=f_0(x_t)+B_t(u_t-f(x_t,c))+w_t.$$ We will add formal justifications for a variety of under-actuated examples in the paper, such as pendulums and drones. To summarize, our model supports not only fully-actuated systems in Eq. (1), but the above dynamics with matched uncertainty. Meta-adaptive control for other under-actuated systems requires something beyond direct disturbance compensation (i.e., it needs some multi-step planning), and is definitely an interesting future direction. We will add a remark in the paper.
> Furthermore, we incorporated a new set of experiments with drones which are suggested by reviewer BB7T, to see how the algorithm scales.
>
> * Thanks to the suggestion, we have conducted the mentioned empirical study of the offline meta-learning algorithm and will incorporate the results into the draft. Our observation is consistent with the claim in the paper and OMAC provides superior performance.

---

> > ### Comment · Reviewer_ZMnJ · 2021-08-31
> > **Thanks for the response**
> >
> > Thanks for the response. I think adding the additional experiments and clarifications on the assumptions on model mismatch in under-actuated settings will sufficiently improve the paper and address my questions. I will update my review accordingly.

---

### Official Review · Reviewer_BB7T · 2021-07-17

**Rating:** 7
**Confidence:** 4

**Summary:**

This paper introduces an integrated representation learning and adaptive control approach named Online Meta-Adaptive Control (OMAC). The approach uses a meta-adaptor to learn a shared representation that can be used across different environments, and uses an inner adaptor to perform environment-specific updates. Non-asymptotic convergence guarantee is provided for the approach. The approach and several baseline methods are evaluated to address a nonlinear pendulum control problem in the experiments.

**Ethical Concerns:**

None.

**Limitations And Societal Impact:**

Several future work topics are briefly mentioned in Section 7. No negative societal impact is identified.

**Main Review:**

The research problem and its challenges are well defined and justified. The proposed approach is novel and has a theoretically sound basis. It can also be applied to various instantiations such as convex, element-wise convex and deep learning cases. The paper is also well organized and well written.

The following comments, mainly on the experiments, may help improve the quality of the paper:

* The research problem in the paper is motivated using several robotic adaptation scenarios; however, the experiments only address a relatively simple pendulum control scenario. Evaluation using a robotics application, e.g., drone control, can make the experiments more relevant and convincing.

* As a follow-up comment, the paper argues the generality of the approach, e.g., it can be used to model fully actuated robotic system. But such statements are not reflected in or supported by the experiments.

* The experiments only provide results obtained by the proposed approach and its simplified versions as baselines (e.g., in Table 5). No comparison with other state-of-the-art adaptive control methods is provided in the experiments. It is difficult to evaluate whether the proposed approach outperforms the current state-of-the-art.

* Providing video demonstrations of pendulum controls by different methods can be helpful for the audience to intuitively evaluate and understand the experimental results.


**Time Spent Reviewing:**

5 hours

---

> ### Author Response · Authors · 2021-08-10
> **Response to Reviewer BB7T**
>
> Thank you for the review and suggestions. We will answer the reviewer’s questions separately as follows:
>
> * Regarding the question about having more robotic applications and video demonstrations, we made a detailed study on drones in changing winds and incorporated the results into the draft. Visualizing the adaptation is also a great idea, we will definitely show pendulum/drone performance in videos, and emphasize how fast different methods adapt.
>
> * Regarding the baselines, as suggested by reviewer ZMnJ, we conducted a comparison against offline meta-learning methods (i.e., first collecting $K$ environments’ data and using a MAML-style regression). Our observation is that OMAC provides superior performance. Please note that most other adaptive control methods, e.g., velocity gradient and composite adaptation, do not directly apply to the multi-environment setting and require extra assumptions, such as a known Lyapunov function.

---

### Review · Ethics_Reviewer_kK8m · 2021-07-22

**Recommendation:**

I suggest the authors briefly discuss these issues in the next version of the paper.

**Ethics Review:**

A couple of the reviewers have flagged minors specific issues - model mismatch and limitations that could easily be addressed in the final version of the paper.

---

### Review · Ethics_Reviewer_5vU4 · 2021-08-09

**Recommendation:** N/A

**Ethics Review:**

I agree with the assessment of the authors and of technical reviewer H6i6 that this work is sufficiently divorced from downstream applications such that the limited discussion of negative social impact in the manuscript is acceptable.

---

### Author Response · Authors · 2021-08-22
**Update: New Results for a Drone**

As suggested by the reviewers, the following table shows the simulation result for a drone with an unknown quadratic air drag model, where the state dimension is 6 and the unknown term $f(x,c)$ is 3-d.

| | no-adapt  |  omniscient | baseline | OMAC (convex) | OMAC (bi-convex) | OMAC (deep learning) |
|---|---|---|---|---|---|---|
| ACE | 1.117 | 0.125 | 0.622 | 0.397 | 0.336 | 0.218 |

---

### Decision · Program_Chairs · 2021-09-28

**Decision:**

Accept (Poster)

**Comment:**

This paper presents an approach to tackle the problem of adaptive nonlinear control over a set of related environments, by learning a shared representation. The approach is sufficiently evaluated and the paper well structured.
This is an interesting paper that would be a positive contribution to the conference.
The authors are advised to make the promised changes.

**Consistency Experiment:**

NeurIPS has a long history of experimentation. In 2014, NeurIPS ran an experiment in which 10% of submissions were reviewed by two independent committees to quantify the randomness in the review process. This year, we repeated a variant of this experiment to see how the quality of the review process has changed over time.  This paper was part of the experiment and was therefore assigned to two committees (consisting of reviewers, an Area Chair, and a Senior Area Chair) that reached independent decisions.  If both committees made the same recommendation, this recommendation was followed. If a single committee recommended acceptance, the paper was accepted (with the exception of a few cases in which the other committee identified what we considered a fatal flaw, e.g., an error in a key result).

This copy’s committee reached the following decision: **Accept (Poster)**

The other committee assigned to the paper recommended **Reject**.  You can find the other set of reviews, along with any follow up discussion with the authors here:
https://openreview.net/forum?id=sGUet7yVVgn